

# Assessment of smoke plume height products derived from multisource satellite observations for wildfire in the western US

Jingting Huang[1,*], S. Marcela Loría-Salazar[2], Min Deng[3], Jaehwa Lee[4,5], Heather A. Holmes[1]

[1] Department of Chemical Engineering, University of Utah, Salt Lake City, 84112 UT, USA
5  [2] School of Meteorology, University of Oklahoma, Norman, 73072 OK, USA
[3] Environmental and Climate Sciences Department, Brookhaven National Laboratory, Upton, 11973 NY, USA
[4] NASA Goddard Space Flight Center, Greenbelt, 20771 MD, USA
[5] Earth System Science Interdisciplinary Center, University of Maryland, College Park, 20740 MD, USA

*Correspondence to:* Jingting Huang (jingting.huang@utah.edu)

10  **Abstract.** As wildfires intensify and fire seasons lengthen across the western U.S., the development of applicable models that can predict the density of smoke plumes and track wildfire-induced air pollution exposures has become critical. Wildfire smoke plume height is a key indicator of the vertical placement of plume mass emitted from wildfire-related aerosol sources in climate and air quality models. With advancements in Earth observation (EO) satellites, spaceborne products for aerosol layer height or plume injection height have recently emerged with increased global-scale spatiotemporal resolution. However, to evaluate column radiative effects and refine satellite algorithms, vertical profiles of regionally representative aerosol data from wildfire emissions need to be measured directly in the field. In this study, we conduct the first comprehensive evaluation of four passive satellite remote sensing techniques specifically designed to retrieve plume height distribution for wildfire smoke. We compare these satellite products with the airborne Wyoming Cloud Lidar (WCL) measurements during the 2018 Biomass Burning Flux Measurements of Trace Gases and Aerosols (BB-FLUX) field campaign in the western U.S. Two definitions, namely "plume top" and "extinction-weighted mean plume height", are used to derive representative heights of wildfire smoke plumes, based on the WCL-retrieved vertical aerosol extinction coefficient profiles. We also perform a comparative analysis of multisource satellite-derived plume height products for wildfire smoke using these two definitions. With the aim to discuss which satellite product is most appropriate under various aerosol loadings and in determining plume height characteristics near a fire-event location or downwind plume rise equivalent height. Our findings highlight the importance of understanding the sensitivity of different passive remote sensing techniques to space-based wildfire smoke plume height observations, in order to resolve ambiguity surrounding the concept of "effective smoke plume height". As additional aerosol-observing satellites are expected to be launched in the coming years, our results will inform future remote sensing missions and EO data selection. This will help bridge the gap between satellite observations and plume rise modeling to further investigate the vertical distribution of wildfire smoke aerosols.

## 1 Introduction

Characterizing the vertical extent of wildfire smoke aerosols near active fire hotspots, also known as plume injection height (PIH) or smoke aerosol layer height (ALH), is a critical task in simulating the long-range transport of wildfire smoke. From a physical perspective, the initial PIH at a fire-event location can be described as the height where the



relatively stable vertical atmospheric layer is located, causing the smoke plumes to accumulate, and the updrafts generated by the plume buoyancy above the fire source to terminate due to turbulence and mixing (Kahn et al., 2007; Labonne et al., 2007; Paugam et al., 2015). Simply put, PIH is commonly viewed as the vertical height to which a buoyant plume core can lift the polluted airmass before smoke plumes begin to bend over horizontally (Raffuse et al., 2012). Practically, plume heights near or downwind of active fire areas are meant to be equivalent to PIH values. It

has been observed that most wildfire smoke plumes move horizontally in single layers through the atmosphere, but some may become stratified into discrete, multiple layers (Mardi et al., 2018; Deng et al., 2022b). However, it is impossible to distinguish aerosol layering at multiple heights without vertically resolved smoke aerosol profiles. Consequently, a single height value is often applied and can be obtained from physics-based numerical models or remote sensing observations. Regardless of whether the vertical structure of wildfire smoke aerosols is homogenous

or heterogenous, a columnar plume height retrieved from satellites is considered a representative ALH. This study focuses on the smoke-specific plumes from wildfires in the western United States (WUS); therefore, we will use smoke plume height ($SPH$) hereinafter to denote this.

Wildfire $SPH$ data observed from space has witnessed a significant advancement in spatiotemporal resolution since the 2000s (Kahn et al., 2007; Ichoku et al., 2012; Lyapustin et al., 2019; Kahn, 2020). Passive satellite sensors are

most widely used to map global wildfire $SPH$ distribution, spanning a substantial range of recently developed techniques and retrieval algorithms. It is important to note that each method to obtain satellite $SPH$ retrievals utilizes a distinct remote sensing technique, resulting in inconsistent definitions of $SPH$. To shed light on their differences, a brief overview of these methods is provided to demonstrate why they yield differing plume height interpretations.

The photogrammetric stereo capability of the Multi-Angle Imaging SpectroRadiometer (MISR) aboard the National

Aeronautics and Space Administration (NASA) Earth Observing System's Terra spacecraft (Diner et al., 1998), combined with the MISR Interactive Explorer (MINX) tool (Nelson et al., 2008, 2013), makes it a practical solution to determine "wind-corrected" $SPH$ values of elevated smoke aerosols. This approach takes into account feature displacements caused by the real motion of plume elements and the stereo parallax shift among different camera views. Another accepted mainstream approach to retrieve $SPH$ information takes advantage of the altitude dependence of

absorption spectroscopic characteristics of molecular oxygen ($O_2$) in the A band at 759–771 nm or the B band at 686–695 nm or the $O_2$–$O_2$ spectral band at 477 nm, which has been successfully applied to a number of passive satellite-supported instruments, including but not limited to POLDER/PARASOL (the POLarization and Directionality of the Earth's Reflectance mounted on the Polarization and Anisotropy of Reflectances for Atmospheric Sciences coupled with Observations from a Lidar platform, Dubuisson et al., 2009), MERIS/ENVISAT (the MEdium Resolution

Imaging Spectrometer installed on the Environmental Satellite, Duforêt et al., 2007; Dubuisson et al., 2009), SCIAMACHY/ENVISAT (the SCanning Imaging Absorption SpectroMeter for Atmospheric CHartographY on board the Environmental Satellite, Corradini and Cervino, 2006; Sanghavi et al., 2012), GOME-2/MetOp (the Global Ozone Monitoring Experiment–2 flying on the Meteorological Operational series of satellites, Sanders et al., 2015; Nanda et al., 2018a; Michailidis et al., 2021), OMI/Aura (the Ozone Monitoring Instrument aboard the Aura spacecraft, Chimot

et al., 2017, 2018), EPIC/DSCOVER (the Earth Polychromatic Imaging Camera loaded on the Deep Space Climate



Observatory, Xu et al., 2019; Lu et al., 2021) and TROPOMI/S-5 P (the TROPOspheric Monitoring Instrument carried on the Copernicus Sentinel-5 Precursor mission, Griffin et al., 2020; Nanda et al., 2020; Chen et al., 2021). Two more spectral channels are sensitive to the vertical distribution and optical properties of highly absorbing non-spherical irregular smoke aerosol particles, namely ultraviolet (UV) and infrared (IR) or thermal bands. These bands could also

be used to pinpoint *SPH* on a global scale. Based on the sensitivity of backward near-UV or UV radiance to the presence of rising absorbing aerosol plumes (e.g., dust and smoke) in a Rayleigh scattering atmosphere (Hsu et al., 1996; Torres et al., 1998; Hsu et al., 1999), previous studies proposed an algorithm called the Aerosol Single-scattering albedo and Height Estimation (ASHE) that jointly retrieves ALH and single scattering albedo (SSA) using UV-aerosol index (UVAI), aerosol optical depth (AOD), and spaceborne lidar backscatter profile from multi-sensor measurements

(Jeong and Hsu, 2008; Lee et al., 2015, 2016). Later, Lee et al., 2020 revised the ASHE algorithm to make it work without the requirement for the lidar backscatter profile in operational environments, of which the *SPH* product is used in this study. More recently, Lyapustin et al., 2019 and Cheeseman et al., 2020 introduced the brightness temperature contrast approach in the Moderate Resolution Imaging Spectroradiometer (MODIS) thermal band (11-μm) for smoke plume identification and characterization. Using this technique, daily *SPH* records on a global

sinusoidal grid have been issued as part of the Multi-Angle Implementation of Atmospheric Corrections (MAIAC) atmospheric product MCD19A2.

When it comes to remotely sensed *SPH* retrievals, passive satellites excel by delivering widespread coverage on a regular basis, all while incurring minimal recurring costs and posing no risks to observers. Yet, dense smoke plumes, cloud cover, or scan gaps between adjoining orbits of sun-synchronous polar satellites can result in unsuccessful fire

detections (Lyapustin et al., 2008). As a complement to these passive retrievals, active spaceborne lidars like CALIOP/CALIPSO (the Cloud-Aerosol Lidar with Orthogonal Polarization aboard the Cloud-Aerosol Lidar and Infrared Pathfinder Satellite Observation satellite, Winker et al., 2009) and CATS/ISS (the Cloud-Aerosol Transport System installed on the International Space Station, McGill et al., 2015) offer high-resolution vertical profiles of aerosol optical signals and enhance the detection of thin smoke layers, though they are bound by a narrow, pencil-like

swath (Kahn et al., 2008). Besides, these remote sensing instruments do not resolve the diurnal variation of wildfire activity, unlike traditional in situ monitoring.

Endeavors to investigate fire behavior and their associated air quality (AQ) have predominantly relied on the use of field data and satellite-based retrievals. Hence, passive and active remote sensing techniques are complementary because of their different observational methods. The deliberate collocation of them provides synergistic insights into

missing pieces of fire information that may not be attainable by either type of technique in isolation (Liu et al., 2019a; Sicard et al., 2019). Unfortunately, in the Intermountain West region of the U.S., there remains a lack of detailed vertical profiles of aerosol optical properties, despite recent field experiments such as Fire Influence on Regional to Global Environments and Air Quality (FIREX-AQ), Western wildfire Experiment for Cloud chemistry, Aerosol absorption and Nitrogen (WE-CAN), Biomass Burning Flux Measurements of Trace Gases and Aerosols (BB-FLUX),

and Fire and Smoke Model Evaluation Experiment (FASMEE). Furthermore, to date, there is no universally accepted methodology for directly deriving wildfire *SPH* from aerosol extinction or backscatter vertical profiles due to the



ambiguous use and definition of the term "effective *SPH*" (Xu et al., 2017). This poses a challenge particularly when one wants to compare columnar *SPH* values from passive remote sensors with actively retrieved three-dimensional (3D) distribution of smoke aerosol vertical structure from active remote sensors.

The primary objective of this study is to address the central research question of which *SPH* definition can effectively interpret a specific satellite *SPH* retrieval algorithm. We introduce two *SPH* definitions built upon vertical profiles of smoke aerosol from airborne lidar data. We then quantify the sensitivity of four passive remote sensing techniques to columnar wildfire *SPH* observations with respect to these two definitions, accounting for the effects of local meteorology, distance from the active fire source, and smoke aerosol loading. Meanwhile, we explore an optimal 115 collocation strategy to compare satellite retrievals with lidar measurements, considering instrument discrepancies in observing *SPH* experimentally. To the best of our knowledge, we present the first comprehensive assessment of multiple satellite-derived wildfire *SPH* products compared with aircraft lidar data. The results of our study clarify the meaning of "effective *SPH*" in the remote sensing and modeling communities, filling a critical gap in uniform plume height comparisons. Our findings also meet the urgent need for a suite of remotely sensed datasets to evaluate the 120 performance of present and future plume dynamic models and smoke modeling frameworks, or to provide inputs to these models that improve the *SPH* characterization required to model the downwind pollutant transport.

## 2 Direct measures of wildfire *SPH*

### 2.1 Satellite-based wildfire *SPH*

The following four space-based wildfire *SPH* retrievals that are quality-controlled will be discussed in our study: (1) 125 new MODIS aerosol products using the MAIAC algorithm (MODIS/MAIAC); (2) a MISR-based global *SPH* database that can be accessed via the MISR Enhanced Research and Lookup Interface (MISR/MERLIN); (3) new VIIRS aerosol products using the ASHE algorithm (VIIRS/ASHE); (4) TROPOMI-based ALH products (TROPOMI/ALH). **Table 1** features further information about these passively remote-sensed *SPH* datasets.

**Table 1: Summary of multisource satellite-derived plume height products.**

| Instrument | Geographic Coverage/ Satellite Orbit | Satellite/Agency | Data Set/Version | Time Period | Resolution |
|---|---|---|---|---|---|
| **MODIS** | global/sun-synchronous polar | Terra, Aqua/NASA | **MAIAC**-derived injection height products/collection 6.1 | February 1, 2000 to present | *horizontal*: 1 km × 1km |
| | | | | | *temporal*: 16-day repeating cycle; one-to-two-day global coverage |
| **MISR** | global/sun-synchronous polar | Terra/NASA | **MERLIN** interface for MISR plume height project /version 2 | 2008–2011 as well as the summers (June, July, August) of 2017 and 2018 | *horizontal*: 1.1 km × 1.1 km |
| | | | | | *temporal*:16-day repeating cycle; 9-day global coverage |



| | | | | | |
|---|---|---|---|---|---|
| **VIIRS** | global/sun-synchronous polar | Suomi NPP/NASA, NOAA | **ASHE**-derived ALH products/research | August of 2013–2018 | *horizontal*: 6 km × 6 km |
| | | | | | *temporal*:16-day repeating cycle; daily global coverage |
| **TROPOMI** | global/sun-synchronous polar | S-5 P/ESA, the Netherlands Space Office, the European Commission | **TROPOMI** level-2 ALH/version 1 | April 30, 2018 to July 1, 2021 | *horizontal*: 3.5 km × 7 km (across x along track) from April 30, 2018 to August 6, 2019; 3.5 km × 5.5 km since August 6, 2019 |
| | | | | | *temporal*:16-day repeating cycle; near-daily global coverage |

### 2.1.1 MODIS/MAIAC

MODIS sensors, located on the Terra (morning sensor, 10:30 AM local solar time) and Aqua (afternoon sensor, 1:30 PM local solar time) satellite platforms and operated in the thermal-IR spectrum, have the unique ability to detect active fires (Salomonson et al., 2002). This twin-MODIS design covers most regions on the equator with at least four observations per day. The number of observations increases as one approaches the poles due to overlapping orbits. Atmospheric properties data created on the MODIS aerosol data using the MAIAC algorithm, known as the MCD19A2 dataset with high resolution (1 km), offers information on near-fire-source aerosol injection height.

By assuming a fixed lapse rate, the MAIAC PIH algorithm utilizes negative thermal contrast at 11 μm between smoke and sufficient neighboring smoke-free pixels and converts the colder brightness temperature into *SPH* estimations (Lyapustin et al., 2019; Cheeseman et al., 2020). The valid range allowed for the MAIAC-based *SPH* is up to 10 km. However, the *SPH* calculation struggles with large smoke areas and small fires emitting low levels of absorbing gases, meaning it requires a high enough plume opacity (AOD at 470 nm ≥ 0.8) to obtain a useful signal. Additionally, when compared to other *SPH* datasets such as MISR and CALIOP, the MAIAC method tends to underestimate the height of smoke plumes, particularly for transporting dilute smoke over time and distance from the source of burning. In spite of these limitations, the MAIAC algorithm provides valuable data within approximately 75–150 km of the identified thermal hotspots for optimal retrieval quality (Loría-Salazar et al., 2021).

### 2.1.2 MISR/MERLIN

With its nine fixed push-broom cameras, MISR aboard NASA's Terra satellite captures images with high precision from nine different angles and four spectral bands, allowing for studies of wildfire and aerosol distributions using stereoscopic techniques, unaffected by bright surfaces (Moroney et al., 2002; Muller et al., 2002). The wealth of data collected by MISR over two decades offers valuable insights into the global climatology of fire in the environment, categorized by various geographic regions, biomes, and seasons (Val Martin et al., 2018; Gonzalez-Alonso et al., 2019). This publicly available database has been used to validate plume rise in models (e.g., Ke et al., 2021) and other satellite-derived datasets (e.g., Lyapustin et al., 2019; Griffin et al., 2020). Recently, a novel interactive visualization tool called MERLIN has been developed to cover applications of the outdated MISR Plume Height Project and



facilitate the exploration and accessibility of over 70,000 records of global wildfire plume data (Boone et al., 2018; Nastan et al., 2022).

MISR's global *SPH* mapping complements aerosol height curtains obtained from spaceborne lidar systems. The blue-band data at 1.1 km horizontal cell size is considered a better choice for capturing the higher injection heights associated with fine smoke aerosols than the corresponding red-band retrievals at 275 m (Nelson et al., 2013). In this

study, we extract blue-band, wind-corrected heights with "good" quality flags. Nevertheless, important lessons can be drawn from the underestimated *SPH* values in the MISR product as follows: (1) the overpass time of MISR in the morning precedes the daytime peak in fire activity, typically in late afternoons when temperatures are highest and relative humidity is lowest; (2) very few coincident overpasses exist over fires during a short time of interest due to the narrow MISR swath, which allows global coverage only approximately once per week. Additionally, the revisit

period of MISR for a specific geographical spot varies from 2 to 9 days, depending on the latitude (Kahn et al., 2007); (3) MISR automated stereoscopic image's dependence on optically distinct plume-like features for accurate height estimation can introduce bias, mainly when dealing with thin smoke or smoke downwind of the active fire source with less defined boundaries (Nelson et al., 2013).

### 2.1.3 VIIRS/ASHE

The launch of subsequent operational VIIRS sensors has been planned for the Joint Polar Satellite System (JPSS) series since 2011, in anticipation of the post-MODIS era (Cao et al., 2013a, 2013b; Goldberg et al., 2013; Wolfe et al., 2013; Wang and Cao, 2019). Owing to a large image swath of 3,040 km and a 12-h global coverage revisiting cycle, mid-latitudes will experience up to 4 looks per day (Wolfe et al. 2013). Even though the VIIRS data has enhanced radiometric measurement quality, a broad spectral range, and a fine spatial resolution (Csiszar et al., 2014;

Schroeder et al., 2014), the 12-h overpass time lag may curtail its efficacy for delineating fire perimeters and assessing fire spread, especially during short fire durations (Cardil et al., 2019).

The research version of the ASHE algorithm (transition to operational processing is underway at the time of writing) provides the plume height of UV-absorbing aerosols like smoke and dust over broad areas, including both near-source and transported plumes (Loría-Salazar et al., 2021). Initially, it leveraged AOD and Ångström Extinction Exponent

(AEE) from the MODIS or VIIRS aerosol product in its retrieval process, as well ALH along the CALIOP track as a constraint (Jeong and Hsu, 2008; Lee et al., 2015). By assuming spatially invariant SSA retrieved along the CALIOP track over a MODIS/VIIRS granule, it has shown the ability to extend the height retrieval beyond the narrow CALIOP track, thereby improving spatiotemporal coverage. This study makes use of a release candidate of ASHE, which can retrieve ALH bypassing the need for CALIOP measurements, benefiting from the synergy between VIIRS and Ozone

Mapping and Profiler Suite - Nadir Mapper (OMPS-NM) (Lee et al., 2020). To further improve its performance, a systematic optimization of the smoke optical models used in the algorithm was carried out by iteratively comparing the retrieved ALH and SSA with those from CALIOP and ground-based measurements offline until satisfactory similarity was found between the results (e.g., Jeong et al., 2022). One simplification made by the algorithm is to assume a single aerosol layer, which may not accurately reflect complex atmospheric conditions characterized by



multiple aerosol layers. Also, its application is limited to UV-absorbing aerosols with moderate to thick optical depths
       (AOD at 550 nm > 0.5–1.0), rendering it ineffective for aerosols with lower optical depths.

### 2.1.4 TROPOMI/ALH

The TROPOMI instrument is the single payload on board of the European Space Agency (ESA) S-5 P satellite mission,
running in the planned timeframe 2017-2024. TROPOMI is a spectrometer that monitors key atmospheric constituents
and aerosol/cloud properties by observing reflected sunlight across the spectral bands in the UV, the visible (270–500
       nm), the near-IR (675–775 nm), and the shortwave IR (2305–2385 nm). Compared to its predecessors (OMI and
       SCIAMACHY), TROPOMI provides high spatially resolved information and is capable of daily global coverage and
       near real-time data, which enables regular monitoring and rapid assessment of changes in the Earth's atmosphere
       (Veefkind et al., 2012).

By analyzing the spectral signature of light that is absorbed by $O_2$ in the A band in the near-IR wavelength range
       between 759 and 770 nm, the TROPOMI ALH algorithm estimates the height of aerosol plumes in the atmosphere
       (Sanders et al., 2012; Nanda et al., 2019). It has shown to be effective in retrieving high plumes up to 8 km in height
       above ground level (AGL), with reduced uncertainties for thicker and lower plumes between 1 and 4.5 km in height
       AGL (Griffin et al., 2020), as well as for dark surfaces (Nanda et al., 2018b). However, it has been found to be biased
low in contrast to other *SPH* datasets such as MISR and CALIOP, most likely due to its tendency to return an
       intermediate plume height when multiple aerosol layers are present (Griffin et al., 2020; Nanda et al., 2020). In
       addition, Nanda et al., 2020 pointed out that cloud contamination would have an impact on the TROPOMI/ALH
       product since it is unable to distinguish between cloud and aerosol signals from the measured radiances. In this study,
       we only use data having a quality assurance value larger than 0.5 to filter mostly cloudy scenes or observations with
geolocation errors.

### 2.2 Airborne lidar measurements

The 2018 BB-FLUX field campaign deployed the upward-pointing Wyoming Cloud Lidar (WCL) on the University
of Wyoming King Air (UWKA) research aircraft that sampled smoke plumes from more than 20 wildfires during 35
flights over the WUS. The airborne WCL measurements of attenuated backscattering coefficient and lidar
depolarization ratio were calibrated on a per-flight basis. The vertical aerosol extinction profiles (units: km$^{-1}$) were
       retrieved with Fernald's method assuming a constant lidar ratio of 60 and evaluated with in situ measurements (see
       Deng et al., 2022a for details). **Table 2** lists the nine wildfire cases during August 2018 used in this paper, and **Fig. 1**
       depicts the matching eleven flight trajectories. The chosen flights are limited to passes of smoke plumes that could be
       attributed to a specific wildfire. Other flights during the campaign are excluded from this study because they target
prescribed fires, small wildfires, clouds, and the air mass containing the aged smoke plumes. We then re-grid valid
       WCL two-dimensional (2D) transects at a vertical resolution of 3 m and a horizontal resolution of about 1.1 km to
       calculate *SPH* and columnar AOD throughout the atmosphere. Compared to satellite observations, the re-gridded
       WCL measurements have a much smaller field of view in the cross-track direction, therefore the WCL can show much
       finer spatial variations in smoke. Moreover, it should be noted that the WCL can be fully attenuated in dense smoke





and unable to detect the actual *SPH*, and the aircraft might fly above the plume bottom height, so the upward-looking WCL only samples partial AOD of the aerosol vertical profiles, which fundamentally differs from that derived from satellite data.

**Table 2: Summary of eleven flight missions during August 2018 that were selected based on the number of collocated pairs between valid lidar transects and satellite overpasses. The flight name because of the wildfire case occurred in the morning**
**is denoted by its date '+ a', otherwise it is denoted by its date '+ b' if it occurred in the afternoon.**

| Flight Date (UTC) | Flight Name | Wildfire Name | Active Fire Location (Latitude, Longitude) | Aircraft Sampling Distance from Active Fire Source (km) | Nth Day After Fire Start Date |
|---|---|---|---|---|---|
| **2018/08/03** | 20180803a | Sharps Fire, ID | 43.467°N, 114.145°W | 18.88 | 6 |
| **2018/08/04** | 20180804b | Sharps Fire, ID | 43.467°N, 114.145°W | 46.50 | 7 |
| **2018/08/08** | 20180808b | Rabbit Foot Fire, ID | 44.856°N, 114.307°W | 21.86 | 7 |
| **2018/08/12** | 20180812a | Rabbit Foot Fire, ID | 44.856°N, 114.307°W | 32.80 | 11 |
| **2018/08/19** | 20180819a, 20180819b | Watson Creek Fire, OR | 42.653°N, 120.818°W | a: 5.26/b: 22.61 | 5 |
| **2018/08/20** | 20180820a, 20180820b | Sheep Creek Fire, NV | 40.773°N, 116.842°W | a: 1.70/b: 1.48 | 3 |
| **2018/08/23** | 20180823a | South Sugarloaf Fire, NV | 41.812°N, 116.324°W | 51.44 | 7 |
| **2018/08/24** | 20180824a | Watson Creek Fire, OR | 42.653°N, 120.818°W | 41.75 | 10 |
| **2018/08/25** | 20180825a | Watson Creek Fire, OR | 42.653°N, 120.818°W | 13.17 | 11 |

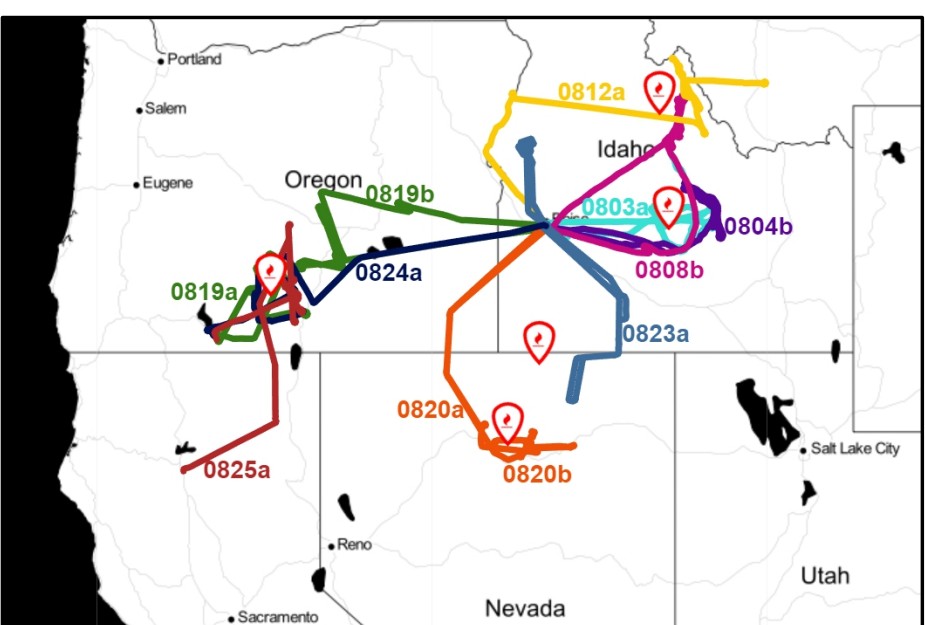

**Figure 1: Color-coded eleven UWKA flight trajectories during the 2018 August BB-FLUX project, each of which was associated with one of nine wildfire cases denoted by fire icons.**



## 3 Methods

### 3.1 Definitions of wildfire *SPH* estimates

The extinction coefficient is a key parameter for the fundamental radiative transfer calculations of wildfire smoke aerosols from the surface to the top of the atmosphere (TOA) (e.g., Ansmann et al., 2018; Solomon et al., 2022) and can yield a linear relation to the particle mass (or volume) concentration (e.g., Mamouri and Ansmann, 2016; Toth et al., 2019; Ansmann et al., 2021). It has been recognized as one of the most frequently observed and reported aerosol optical properties to characterize the atmospheric vertical structure and establish a height retrieval algorithm in previous studies (Gordon, 1997; Dubovik et al., 2011; Sanghavi et al., 2012; Hollstein and Fischer, 2014; Ding et al., 2016; Wu et al., 2016; Xu et al., 2017). Lidar-based active remote sensing technology usually delivers attenuated backscatter signal intensity that is then processed to produce vertical profiling of the aerosol extinction coefficient by designating an extinction-to-backscatter ratio (Liu et al., 2015; Rosati et al., 2016; Baars et al., 2021).

Two definitions have been proposed and widely used to derive a representative height of wildfire smoke plumes based on the vertical distribution of aerosol extinction coefficient at a given spectral wavelength from active lidar measurements. The concept of "effective *SPH*" can be defined either through smoke aerosol layer boundaries or by considering the complete vertical profile. One identifies the topmost height of the plumes according to the geometric boundary of the aerosol layers. Another is deduced from the average height of the aerosol layers weighted by the extinction (or backscatter) coefficient, indicating the radiative properties of wildfire smoke particles. In this section, we will present a detailed explanation of these two definitions and apply them to the full vertical profiles of WCL-measured aerosol extinction coefficient within the troposphere. The height hereinafter is computed in kilometers AGL.

### 3.1.1 Plume top (*SPH*<sup>top</sup>)

This definition is built on the wavelet covariance transform (WCT) approach given by Gamage and Hagelberg (1993), which is an automatic algorithmic process to extract geometrical features of interest. Since it can detect the aerosol layer locations of subtle but coherent transitions according to their strength and sign, the WCT analysis has been applied to detect realistic high-resolution atmospheric structures at a variety of vertical spatial scales, such as a well-mixed convective boundary layer top (e.g., Brooks, 2003; Baars et al., 2008) and the edges of lofted aerosol layers (e.g., Davis et al., 2000; Siomos et al., 2017). Here we only focus on the derivation of the top of smoke plumes for wildfire, referred to as $SPH^{top}$.

The WCT method is expressed as

$$W_f(a,b) = a^{-1} \int_{z_b}^{z_t} f(z) h\left(\frac{z-b}{a}\right) dz, \tag{1}$$

with a step function, the so-called Haar wavelet $h_{a,b}(z)$, which can be defined as



$$h\left(\frac{z-b}{a}\right) = \begin{cases} +1: b - \dfrac{a}{2} \leq z \leq b \\ -1: b \leq z \leq b + \dfrac{a}{2} \\ \phantom{-}0: \quad elsewhere. \end{cases} \tag{2}$$

In **Eq. (1)**, $f(z)$ is the lidar signal of interest as a function of height AGL, $z$ (in our case the aerosol extinction profile $\beta(z)$ at 355 nm), and $z_t$ and $z_b$ are the upper and lower limits of the profile. For any arbitrary element of the Haar basis $h_{a,b}(z)$ as shown in **Eq. (2)**, $a$ is the dilation parameter in relation to the spatial spectrum of the function, and $b$ is the translation parameter indicating the location at which the function is centered, respectively.

The local match or similarity between the Haar wavelet $h_{a,b}(z)$ and the lidar extinction signal $\beta(z)$ is measured in the
covariance transform $W_f(a,b)$, which can be interpreted as a pattern search for a sudden jump. Accordingly, the position of the local maxima (i.e., positive peaks) in the return WCT signal approximately marks the layer top; likewise, the position of the local minima (i.e., negative peaks) of the covariance transform $W_f(a,b)$ roughly coincides with the layer bottom. To put it in another way, the identification of strong variations in the vertical gradient of the aerosol extinction profile $\beta(z)$ is useful to locate the boundaries between aerosol layers. Inspired by Michailidis et al. (2021,
2023), we report the correct location of *SPH^top* by adopting the last positive peak in the corresponding WCT profile from the surface to the upper atmosphere if some physical-related constraints are satisfied. The optimum value for $a$ affects the number of sufficiently thick aerosol layers that can be retrieved successfully. We therefore limit the minimum acceptable wavelet dilation $a$ to be equal to 54 times the vertical resolution of the aerosol extinction profile $\beta(z)$ at 355 nm, i.e., $a$ = 162 m used in this study. To filter noise in the return WCT signal, a minimum threshold
value is set to 0.05. The values of *SPH^top* are clearly extracted using this approach both for a single-layered and multi-layered aerosol structure as illustrated in **Figs. S1** and **S2**, respectively.

### 3.1.2 Extinction-weighted mean plume height (*SPH^ext*)

Given an aerosol extinction coefficient profile $\beta(z)$ with $n$ lidar vertical levels, this definition weighs each height AGL interval $z_i$ (in our case $z_i$=3 m) for $i$-th level with the height-dependent extinction coefficient $\beta(z_i)$ as described
in Koffi et al. (2012), and then calculates the weighted mean height (i.e., *SPH^ext*) as follows:

$$SPH^{ext} = \frac{\sum_{i=1}^{n} \beta(z_i) \cdot z_i}{\sum_{i=1}^{n} \beta(z_i)}. \tag{3}$$

The above derivation (**Eq. (3)**) has been widely applied in previous literature and considered ideal for comparisons with the ALH retrieval from passive satellite sensors (Chimot et al., 2018; Kylling et al., 2018; Liu et al., 2019b; Nanda et al., 2020), since it offers a simple and useful means to represent the aerosol vertical distribution as a single
height value. For example, in some cases where a single and homogenous (i.e., same particle size and optical properties) aerosol layer is found in the atmosphere, *SPH^ext* gives an indication of the aerosol layer's center of mass. However, when it comes to a complicated vertical structure of multilayer aerosols, *SPH^ext* may be observed at a vertical level with minimal smoke aerosol loading because smoke plumes are present at multiple heights.





**3.2 Lidar-satellite collocation method**

Even at close range to the source and in a short amount of time, the vertical extent of wildfire plumes can vary substantially. This is because specific vegetation types and fuel structures, terrain characteristics, or ambient meteorological conditions during atmospheric transport processes are more favorable to aerosol aging mechanisms and plume rise behaviors than others (Paugam et al., 2016; Junghenn Noyes et al., 2022). Passive satellite remote sensing of wildfire *SPH* provides an indirect measure of columnar quantities at a relatively coarse spatial resolution,

representing the spatial average of a highly variable pixel area of fire activity and therefore smoke plume behavior. Active airborne lidar instead collects instantaneous vertical segments of smoke aerosols only along its flight path, which in turn lacks large-scale spatial representation. This fact precludes any perfect match between aircraft observations and satellite retrievals. The disparity in sampling time between airplanes and satellites for the same cluster of wildfire plumes, on the order of minutes to days, presents another inherent challenge and thus yields few

perfectly matched pairs. To make proper comparisons between space- and aircraft-based observing platforms, much deliberation is required in determining the time interval and the distance for collocation pairs of satellite observations and lidar measurements (Junghenn Noyes et al., 2020).

In our study, airborne lidar measurements are integrated to an along-track spatial resolution of about 1.1 km and thoroughly cloud-screened. Amongst all the satellite-derived *SPH* products as discussed above, the VIIRS/ASHE has

the largest pixel size (6 km × 6 km) across the longest period (a nominal temporal duration of 6 minutes). Hence, to ensure that adequate collocation pairs are available within one half hour in response to high-level wildfire smoke plume activity, we decide to use a sampling time window of 12 minutes that corresponds to twice the maximum time span of an orbital swath (one scene) in the multi-sensor satellite data. Broadly, we come up with two methods to collocate our aircraft observations with four satellite products in connection with the spatial statistics of the satellite

*SPH* data. One using an average of the surrounding satellite pixels of a lidar point (hereinafter called the "spatial averaging method") and another using a nearest neighbor search to create a matched pair of lidar-satellite observations (hereinafter called the "matched pair method"). For both methods, we first assume that horizonal changes in wildfire smoke plume spread area are negligible during short time intervals. Moreover, there's a chance that multiple satellite pixels coexist in proximity to a single lidar point when satellite orbits and flight legs intersect. Most importantly, each

satellite dataset maintains its native resolution rather than being resampled to a uniform grid for all products. This decision may bring about the use of a different collocation method for each satellite data set to showcase its unique characteristics.

The first collocation method, i.e., the spatial averaging method, calculates an averaged value of the satellite retrievals within an area of a fixed search radius around the lidar measurement. For MODIS/MAIAC and MISR/MERLIN, since

they have the finest spatial resolution (described in **Table 1**), it predisposes them to have multiple collocations inside a circular area of a given search radius centered on the lidar point. The nature of their noise is smoothed by taking the average of all satellite retrievals in this circular area for a given sampling time, which is a common practice in the remote sensing field (e.g., Virtanen et al., 2018). Also, considering that there are fewer collocated satellite retrievals of the coarse resolution within the search area, such as VIIRS/ASHE and TROPOMI/ALH, we apply our second





collocation method, i.e., the matched pair method. This method is more sensitive to the location of a single satellite pixel coinciding with each point-like airborne lidar measurement. The closest satellite pixel to the nearby lidar point within the given sampling distance and time window is chosen for each match.

The main sources of collocation mismatch uncertainties are: (1) misalignment between the satellite pixel size and the lidar observation point; (2) wind-driven advection (e.g., a high fire-induced horizontal wind can reach the maximum

value of 10 m s$^{-1}$ (Liu et al., 2019c), which can displace fire-related smoke aerosols 3.6 km in 6 minutes); (3) intrinsic positioning and navigation errors. To investigate the effect of the search radius size for the two collocation methods, we use 20 sampling distances ranging from 1 km to 20 km for the radius of the circular region, by assuming a worst-case windy environment of 30 m s$^{-1}$ which results in wildfire smoke aerosol layers migrating ~20 km during the maximum allowed time-interval between observations of 12 minutes in our coincidence criteria. Local *SPH* spatial

variability over scales up to ~20 km can introduce uncertainty in *SPH* comparisons. Accordingly, the standard deviation (*STD*) of the multi-sensor satellite *SPH* retrievals around a lidar point (denoted by $\sigma_{SPH}$) is calculated and plotted in **Fig. S3** to assess the representativeness of not only the point-like lidar observation for the search area covered by the satellite data using the spatially averaging method but also the closest satellite value within the selected spatial criteria using the matched pair method. With increasing distances, all *STD* curves for the satellite retrieved

*SPH* display asymptotic behavior. These values can be interpreted as an upper limit of the *SPH* errors owing to our method of collocation. It is important to optimize the inclusion criteria for the lidar-satellite comparison. For example, a low number of nearby satellite pixel counts shows higher spatial sampling uncertainty, and a low number of one-to-one collocation pairs indicates weaker statistics in calculating the *STD*. While calculating the mean *STD*, $\overline{\sigma_{SPH}}$ from all collocations the average number of nearby satellite pixels within a searching radius per collocation and the total

number of one-to-one collocation pairs are also recorded and can be used as thresholds. The best search radius is thus set to 6 km. A collocated satellite *SPH* can be taken to be generally representative of *SPH* values of all other observations within a 6 km radius circle centered around the WCL data point, with an average *STD*-calculated uncertainty ($\overline{\sigma_{SPH}}$) of ~220 m for MODIS-Terra/MAIAC, ~173 m for MAIAC-Aqua/MAIAC, ~258 m for MISR/MERLIN, ~300 m for VIIRS/ASHE, and ~152 m for TROPOMI/ALH.

**3.3 Reconstructed lidar vertical cross-sections**

During flight legs perpendicular to the mean wind, the UWKA samples the fire plume along nearly the same flight track, albeit at different heights. The UWKA operates at a cruise speed of approximately 90 m s$^{-1}$, enabling it to capture data from different altitudes and angles. As mentioned earlier, the WCL system uses laser beams to measure the optical properties of the plume, but we recognize that WCL is limited in its ability to penetrate and sample dense smoke.

Therefore, the lidar can only provide a partial vertical segment of the fire plumes' cross-section, particularly from the lowest flight altitude.

We reconstruct the vertical structure of wildfire smoke plumes using consecutive WCL transects from different flight legs. This post-processing approach presents a more comprehensive view of pseudo-vertical profiles of the aerosol extinction coefficient, and thus provides valuable reference lidar-determined *SPH* data for robust analysis. The



reconstruction process involves several key steps following Deng et al. (2022b):

I.  Applying extinction coefficient threshold: WCL transects are collected from eleven flight tracks with valid collocation pairs. To separate densely localized fresh smoke from the aged background smoke, an extinction coefficient threshold of 0.1 km$^{-1}$ is applied. This step helps remove background noise and signal attenuation in the WCL data and ensures a clear distinction between different smoke components.

II.  Manual identification of flight legs sampling the same fire plumes: we select flight legs sampling the same fire plumes, by examining flight track maps and locating the areas where multiple flight legs intersected with the fire plumes.

III.  Interpolating discontinuous flight segments to a complete vertical cross-section: to display the vertical cross-section of the smoke plumes more smoothly and to aid further analysis and interpretation,
scattered lidar points with extra 2D vertical structure information from various flight legs are interpolated to form a continuous line. The interpolation process relies on the fact that the change in latitude or longitude of the flight track is monotonic.

## 4 Results and discussion

### 4.1 Leveraging airborne lidar measurements to characterize plume behavior and *SPH*

Our study attempts to determine a reliable method for evaluating the effectiveness of a satellite remote sensing technique in retrieving *SPH* data from wildfires using airborne lidar measurements. It is worth noting that the lidar profiles allow for multiple aerosol layers to be sampled. However, the conventional passive satellite aerosol height retrieval algorithm assumes the presence of a single, homogeneously distributed aerosol layer throughout the entire atmosphere. Despite different measurement concepts when it comes to multiple layers of plumes, to ensure
comparability between passive and active observations of wildfire plume behavior and ease of calculation, we emphasize the significance of an effective height parameter. The two *SPH* definitions used to determine this parameter give an indication of the height of the wildfire smoke aerosol distribution as a single number.





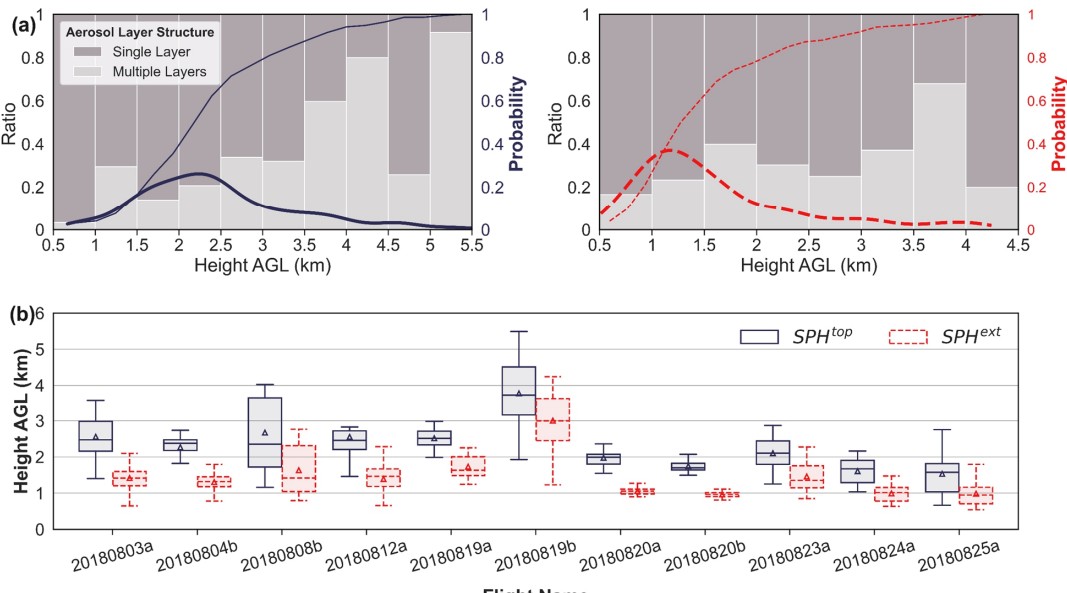

**Figure 2: (a) The ratios of single-layered and multi-layered aerosol structures using two different WCL-determined *SPH* definitions, i.e., *SPH^top* (left, blue solid) versus *SPH^ext* (right, red dotted), accompanied by the probability mass function (PMF; thick line) and cumulative distribution function (CDF; thin line) curves. Note that the WCL plume height data are equally binned by setting the bin-width parameter to 0.5 km, and the bins are spread out in the range from 0.5 km to 5.5 km for *SPH^top* and in the range from 0.5 km to 4.5 km for *SPH^ext*. (b) WCL-determined *SPH* definition comparisons for eleven BB-FLUX flight missions presented by box plots. Upper and lower whiskers represent the 95th and 5th percentiles, respectively, while the box spans from the 25th percentile to the 75th percentile. The line inside the box represents the median (the 50th percentile), and the triangle indicates the mean of the range of plume height values.**

In **Fig. 2a**, the height distributions of wildfire smoke plumes are shown using two different *SPH* definitions during BB-FLUX in August 2018. One should be cautious in identifying key criteria used to define *SPH* prior to assessing the satellite retrievals. This is because *SPH^top* has a vertical extent spanning from 0.5 km to 5.5 km, with the most common height being approximately 2.25 km. On the other hand, *SPH^ext* exhibits a vertical range from 0.5 km to 4.5 km, with its peak observed at roughly 1.2 km. At *SPH* values less than 3.5 km, the occurrence of smoke plumes identified within one single layer is significantly higher than that of multi-layered smoke plumes (> 60% for each height bin), suggesting that the columnar *SPH* values obtained from satellite retrievals can be compared with those measured via lidar profiles, as smoke plumes produced by wildfire typically exhibit a single aerosol layer structure. This finding holds particularly true for wildfires of decreased fire intensity under suppression operations (i.e., the date approaches the fire contained date in **Table 3**), as investigated in this study. In **Fig. 2b**, there is no clear single pattern for the vertical profile of the smoke plume due to the coupled interactions between the fire and atmosphere. Various characteristics, such as weather conditions and fuel types, can influence this relationship. Another critical factor is how far the airborne lidar is from the center of wildfire activity, which will be addressed in **Sect. 4.4**. The aforementioned results have implications for models and retrieval algorithms that presume a standard atmospheric vertical profile with a fixed temperature lapse rate and smoke concentrations. However, the upward-sampled WCL can only provide a partial vertical segment and not a fully resolved cross-section of the smoke plumes from the lowest



flight height due to the restricted lidar laser penetration in optically thick smoke plumes. Therefore, the results presented here underestimate the $SPH^{top}$ (i.e., optically thick plumes limiting vertical extent) and overestimate the

$SPH^{ext}$ (i.e., the upward-pointing lidar not sampling below aircraft). It should be noted that the range of WCL $SPH$ measurements for both morning and afternoon flight missions on August 20, 2018, is limited because only a small fraction of flight tracks are considered valid transects for reconstruction.

**Table 3: Summary of nine wildfire cases with its general information with respect to start date, approximate contained date, and approximate acres burned, which were collected from the GeoMAC (Geospatial Multi-Agency Coordination**
**Group) historic fire dataset (last access: 8 June 2023).**

| Wildfire Name | Start Date | Approximate Containment Date | Approximate Acres Burned |
|---|---|---|---|
| **Sharps Fire, ID** | Jul 29, 2018 | Aug 12, 2018 | 64, 812 |
| **Rabbit Foot Fire, ID** | Aug 2, 2018 | Sep 19, 2018 | 36, 031 |
| **Watson Creek Fire, OR** | Aug 15, 2018 | Sep 9, 2018 | 59, 067 |
| **Sheep Creek Fire, NV** | Aug 18, 2018 | Aug 23, 2018 | 59, 789 |
| **South Sugarloaf Fire, NV** | Aug 17, 2018 | Sep 3, 2018 | 233, 608 |

How smoke aerosols are vertically distributed throughout the atmosphere plays a dominant role in AOD-based surface particulate matter (PM) models. High-elevation smoke aerosol layers above the planetary boundary layer height ($PBLH$) lead to high column AOD while not elevating the near-surface PM levels. Generally, aerosol concentrations are low in the higher, relatively stable atmospheric layers above the planetary boundary layer (PBL). However, a large

wildfire, usually defined as a fire area greater than 1,000 acres burning in the WUS (Linley et al., 2022), tends to have vigorous buoyant plume cores that can lift the smoke plumes to the free troposphere (FT) and even the stratosphere under favorable atmospheric conditions. Therefore, the ratio of effective $SPH$ to $PBLH$ ($SPH:PBLH$) is a better indicator of the AOD and surface PM concentration relationship. Now if we incorporate the modeled $PBLH$ from the Weather Research and Forecasting Model (WRF) as indicated in **Fig. 3**, we can better understand local meteorology

and its impact on wildfire $SPH$. The WRF model for our use has an outer domain extending over the WUS. with a 4 km spatial resolution, nudged with observations from weather stations as well as balloon soundings. $PBLH$ is recalculated from the WRF simulations using the vertical potential temperature gradient method or the Richardson number method. The locations and elevations of each balloon sounding station are in **Table S1**, and details of the WRF model configuration are in **Table S2**. The results of the WRF model evaluation are in **Fig. S4**.

Based on the specific wildfire information in **Table 3**, we can qualitatively discuss the differences between $SPH^{top}:PBLH$ and $SPH^{ext}:PBLH$ for each wildfire in terms of their start dates, approximate containment dates, and approximate acres burned. The ratio $SPH:PBLH$ can explain a joint interaction between buoyant plume cores and complicated boundary layer mixing (e.g., entrainment and wind shear). It also depends on other important factors such as the fire size, distance from the fire source, and the fire spread. In some cases, high $SPH^{top}:PBLH$ (> 1) but low

$SPH^{ext}:PBLH$ (< 1) occur concurrently, as shown in **Fig. 3**. This means that a higher columnar AOD does not necessarily give rise to the majority of the smoke plume concentrations being above the PBL. For instance, the Watson



Creek Fire that started on August 15, 2018, had two flight missions, 0819a and 0819b, and their aviation operation dates were fairly close to the fire start date compared to 0824a and 0825a. The challenging terrain with dense fuel on the ground facilitated rapid fire spread, and no containment efforts were in place. Therefore, we can expect that the

intense fire behavior would generate a higher amount of smoke plumes injected into the FT, where both $SPH^{top}$:$PBLH$ and $SPH^{ext}$:$PBLH$ are larger than 1. Five days later, as the fire activity reduced and containment of the fire increased to 15%, there was likely more smoldering and thus lower plume heights. $SPH^{ext}$ reaches a similar level to the $PBLH$, although $SPH^{top}$:$PBLH$ remains relatively high. When comparing the morning and afternoon $SPH$ patterns, the morning $SPH$ relationships might be less complex and potentially easier to model. Basically, turbulence, convection,

and fire-atmosphere interactions contribute to more chaotic plume and PBL dynamics in the afternoon, causing the growth rate of the fire to exceed the growth rate of the PBL. The Sheep Creek Fire is an exception. It was accidentally begun by a crashed helicopter but was nearly 100% suppressed within one week, due to a timely and consistent fire response making rare $SPH$ behavior in the afternoon possible, where smoke plumes reside within the PBL. Additionally, a significant portion of the lidar vertical cross-section is missing for the 0823a flight during the South

Sugarloaf Fire, as depicted in **Fig. S6g**. In spite of the fire's high severity, which categorizes it as an extreme wildfire episode, the absence of the extinction coefficient data as well as lidar measurements in the downwind region (described in **Table 2**) leads to inaccurate, low estimates of $SPH^{top}$ and $SPH^{ext}$.





**Figure 3: Comparison of the 30-min average *PBLH* obtained from WRF simulations (grey bars) with the WCL-determined *SPH* using two different definitions (*SPH^top*, blue bars; *SPH^ext*, red bars) for the morning (left) and afternoon (right) flight missions. Note that the height of the bar in the bar charts represents the mean of the range of plume height values, and the length of the horizontal error bars displays the *STD*-calculated uncertainty.**

## 4.2 Reconstructed lidar curtain and lidar-satellite collocation

It is necessary to implement post-processing procedures to conduct a comparative analysis between the lidar observations and the satellite retrievals. Here we present detailed reconstructed lidar vertical cross-sections of aerosol extinction coefficient along with collocated satellite *SPH* data on August 19, 2018, for the morning (0819a, **Fig. 4**)



and afternoon (0819b, **Fig. 5**) flight missions. Similar plots are included in **Figs. S5** and **S6** for each flight. **Figs. 4a** and **5a** demonstrate that the smoke plume coverage of the MISR/MERLIN product aligns well with the manually identified plume area and reveals high-resolution *SPH* retrievals. In contrast, the MODIS/MAIAC product with the
highest spatial resolution displays lower *SPH* values in general over primary biomass-burning regions. Meanwhile, both the VIIRS/ASHE product and the TROPOMI/ALH product indicate that higher *SPH* values are generally shifted towards the downwind region of fire sources.

The vertical distributions of wildfire smoke aerosols (**Figs. 4b** and **5b**) are useful to visualize the smoke plume structure and provide more information about the physical processes influencing aerosol layering in the atmosphere.
A visual comparison of the *SPH* values from the four satellite products and the WCL is presented in **Figs. 4b** and **5b**. In **Fig. 4b**, even when faced with intricate aerosol structures, the MISR/MERLIN data is capable of reaching $SPH^{top}$, except for thin plumes with comparably low AOD values. The MODIS-Terra/MAIAC data appears similar to $SPH^{ext}$, although it is unable to distinguish multiple aerosol layers and consequently produces exceptionally low *SPH* values. Since the reconstructed aerosol vertical cross-section for **Fig. 5b** is located in the downwind region of the burn area,
there is an increase in $SPH^{top}$ and $SPH^{ext}$ as the distance from the fire source increases. It is not advisable to use the MODIS-Aqua/MAIAC product for estimating downwind *SPH* due to its suboptimal performance in such scenarios. Regarding heterogeneous aerosol vertical profiles, the spatial agreement between the collocated VIIRS/ASHE *SPH* values and two *SPH* definitions' general trends is poor, despite achieving, on average, a good numerical agreement with $SPH^{top}$. This is partly due to the coarse spatial resolution of OMPS UVAI data used in the algorithm (~50 km at
nadir; ~100 km near the scan edge) not being able to represent finer-scale features. The TROPOMI/ALH data seems consistent with the valid $SPH^{ext}$ values, given the observed overestimation of $SPH^{ext}$ attributable to the elevated flight height. The potential explanation for this phenomenon is that in cases where there may be several layers of smoke aerosols, the retrieved *SPH* would be the average height of the plume much lower than where the optically thick aerosol layer is placed (Michailidis et al., 2023).

According to these results and specific fires studied, the MODIS/MAIAC product struggles with most heterogeneous aerosol structures even in close proximity to active fire sources. The evaluation of the MODIS/MAIAC-derived *SPH* in the afternoon is lacking in literature because MISR onboard the Terra as a comparison data set does not coincide with the Aqua overpass time in the afternoon. Even though the MISR/MERLIN product aims to capture the top boundary of the smoke plume, it can be highly biased in thin plume height estimates with low AOD or a more complex
aerosol structure with multiple aerosol layers. The challenges observed for the VIIRS/ASHE retrievals are: (1) poor correlation with general trends in lidar measurements; (2) it may not accurately represent complex atmospheric conditions with multiple aerosol layers. Among the four satellite *SPH* datasets, TROPOMI/ALH has the lowest *SPH* variance, which is not ideal for application purposes as the real-world wildfire activity varies significantly across spatial areas. However, elevated smoke layers with a high aerosol loading, over dark surfaces at not very high altitudes
are favorable for the TROPOMI ALH algorithm to retrieve vertically localized aerosol layers in the free troposphere.

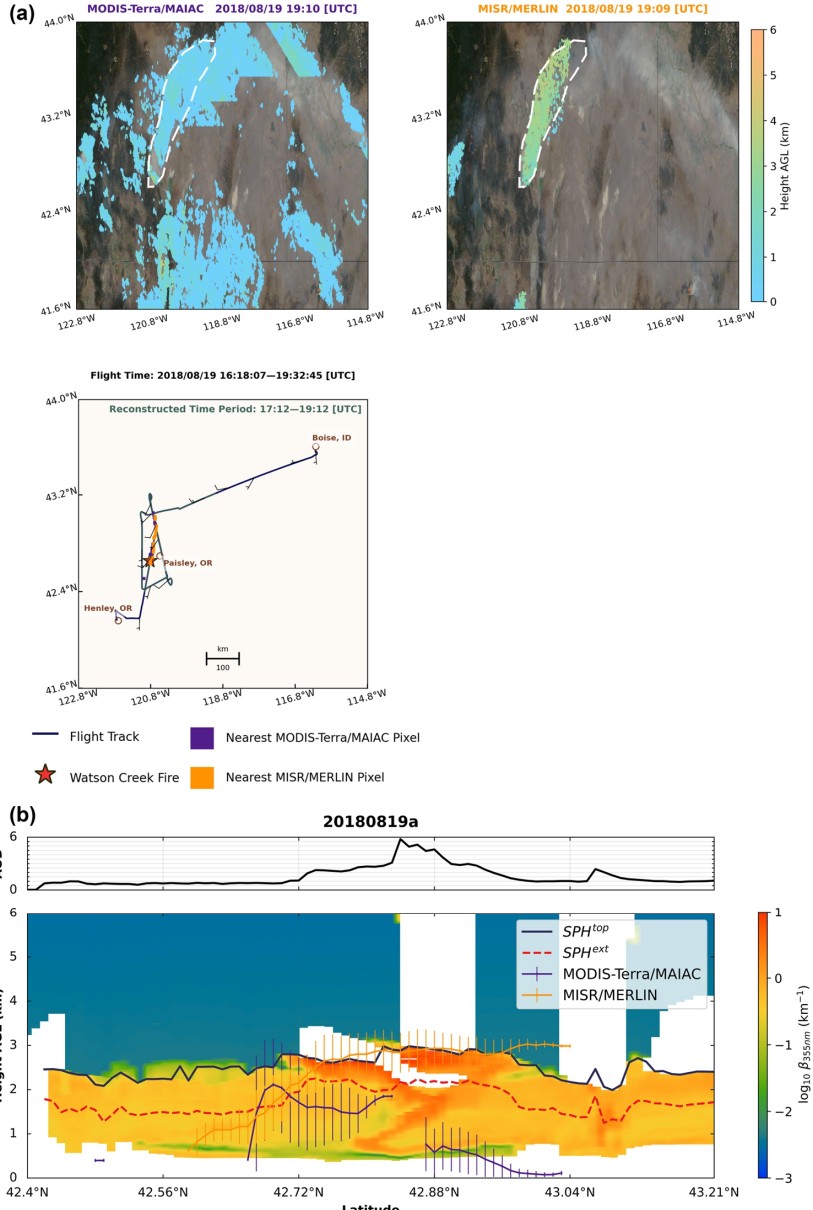

Figure 4: (a) The nearest satellite pixels (MODIS-Terra/MAIAC; purple, MISR/MERLIN: orange) to the corresponding airborne lidar points along the 20180819a flight track during the reconstructed time period from 17:12 to 19:12 UTC highlighted in green. The star symbol indicates the center location of the Watson Creek fire taken from the incident report system (Inci-Web). The NASA WorldView MODIS Terra true-color (i.e., corrected reflectance) images are shown alongside the satellite-retrieved *SPH* maps with the user-drawn smoke plume polygons (denoted as the dashed white region). (b) (Bottom panel) Composite latitude-height cross-sections of the reconstructed WCL vertical aerosol extinction coefficient, overlaid with performance comparisons for variations of WCL-determined *SPH^top* and *SPH^ext* as well as the collocated satellite-retrieved mean *SPH* with error bars for the Watson Creek fire in the morning on August 19, 2018; (top panel) the corresponding AOD variations at 355 nm.

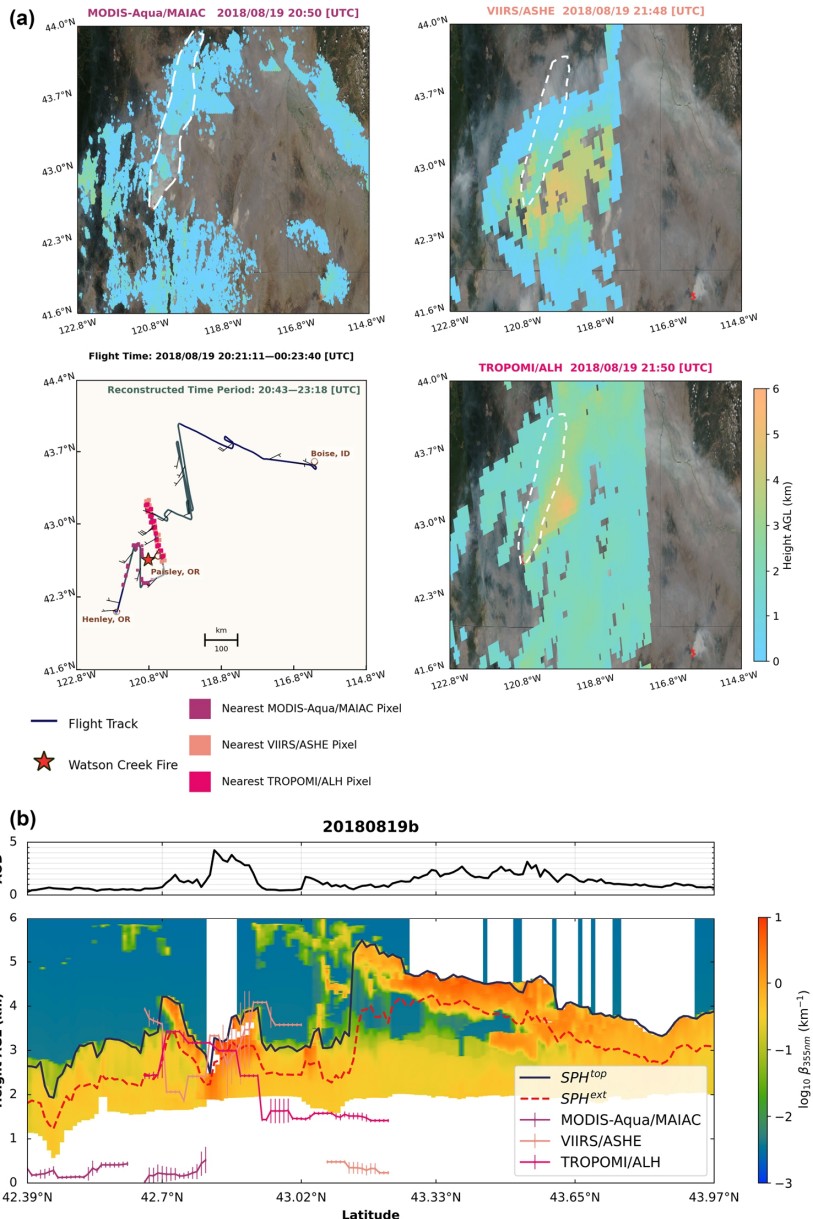

**Figure 5: (a)** The nearest satellite pixels (MODIS-Aqua/MAIAC, violet; VIIRS/ASHE, pink; TROPOMI/ALH, magenta) to the corresponding airborne lidar points along the 20180819a flight track during the reconstructed time period from 20:43 to 23:18 UTC highlighted in green. The star symbol indicates the center location of the Watson Creek fire taken from Inci-Web. The NASA WorldView true-color images (both MODIS Aqua and VIIRS are used) at the corresponding moment are shown alongside the satellite-retrieved *SPH* maps with the user-drawn smoke plume polygons (denoted as the dashed white region). **(b)** (Bottom panel) Composite latitude-height cross-sections of the reconstructed WCL vertical aerosol extinction coefficient, overlaid with performance comparisons for variations of WCL-determined *SPH^top* and *SPH^ext* as well as the collocated satellite-retrieved mean *SPH* with error bars for the Watson Creek fire in the afternoon on August 19, 2018; (top panel) the corresponding AOD variations at 355 nm.



**4.3 Quantitative evaluation of satellite-derived *SPH* interpreted from airborne lidar data**

**Table 4** summarizes the statistical evaluation for collocated pairs for wildfire *SPH* between multiple satellite products and lidar measurements, where *MB* (km) = 0, *MAE* (km) = 0, *RMSE* (km) = 0, and *r* (unitless) =1 indicate perfect agreement. The set of evaluation metrics can be calculated by Eqs. (A1) to (A4). The resulting *SPH* values used to

calculate evaluation metrics are the means of all successful collocations using reconstructed lidar vertical cross-sections. It should be noted that **Table 4** is the combination of **Tables S3** and **S4**, using spatial averaging and matched pair methods for collocation relating to different satellite products.

The MAIAC PIH algorithm has low confidence in *SPH* retrievals compared to the WCL *SPH* measurements using two definitions, especially in the afternoon. One reason might be that the MAIAC algorithm cannot achieve strong

negative thermal contrast, that is, the smoke pixel is not enough "colder" than the background in the afternoon when the fire activity is most active. Moreover, assuming an average lapse rate over mountainous terrains instead of more accurate atmospheric temperature profiles from reanalysis data can introduce more inherent uncertainties in *SPH* estimates. A more significant difference between the MODIS/MAIAC product and the definition of $SPH^{top}$ is found compared to the definition of $SPH^{ext}$, indicating the limitation of high enough total AOD to ensure sufficient gaseous

absorption constrains its ability to detect $SPH^{top}$. Therefore, applying the definition of $SPH^{ext}$ to evaluate the MODIS/MAIAC product would be recommended.

The MISR/MERLIN data fluctuates from 0.625 km to 3.029 km, and the corresponding $SPH^{top}$ determined by lidar profiles varies from 1.254 km to 2.982 km. The mean, *STD* and quartiles of the collocated MISR/MERLIN data exhibit relatively small biases. The MISR/MERLIN product outperforms the other three datasets for capturing $SPH^{top}$ as seen

from the lowest values of *MB*, *MAE*, and *RMSE*. It also has a relatively moderate positive relationship (*r* = 0.551) with the spatial changes in wildfire $SPH^{top}$. This is in line with what we anticipated as contrast features are visible inside plumes and between smoke aerosols and the terrain surface through multiple, angular views, allowing the MISR stereo technique to capture the evolution of wildfire smoke plumes in the cases studied.

Lee et al. (2015) highlighted that the VIIRS/ASHE product performs well over mountainous areas due to the surface

elevation consideration during the retrieval process. Although the mean values and general distribution of both satellite retrievals and lidar observations are fairly close, the VIIRS/ASHE data has a wider spread of values (larger *STD*), a slight tendency to underestimate the $SPH^{top}$ by nearly 116 m, and lower plume height extremes (maximum and minimum plume heights). A fraction within 1.5 km of 14% for VIIRS/ASHE *SPH* leads to some outliers, which are reflected in higher *MAE* and *RMSE*. These large outlier errors could be attributed to difficulties for passive sensors in

measuring the presence of multi-layered aerosols (see **Figs. 5b** and **S6d**), and a potential high AOD bias over bright surfaces. However, the negative spatial correlation (*r* = -0.22) between the VIIRS/ASHE data and the WCL-determined *SPH* suggests significant discrepancies in their spatial resolution when collocating.

On the definition of $SPH^{ext}$, the TROPOMI/ALH product slightly overestimates *SPH* by approximately 158 m, but maintains overall reasonable performance as indicated by the *MAE* and *RMSE* values, and a weak positive correlation

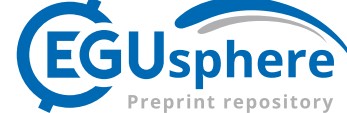

($r$ = 0.241) with lidar observations. However, this evaluation could be influenced by limited collocations. Choosing the appropriate *SPH* definition to interpret the TROPOMI ALH algorithm depends case by case as shown in the reconstructed lidar curtains (**Sect. 4.2**). For multi-layered structures detected in the sample cases (**Fig. 5b**), the *SPH* output from the TROPOMI ALH algorithm is regarded as the average computation of aerosol optical properties. On the other hand, the TROPOMI ALH algorithm shows encouraging potential for characterizing $SPH^{top}$ in homogenous

well-developed smoke layers (**Figs. S6f** and **S6i**). When using the TROPOMI/ALH product, multilayered aerosols, inaccurate aerosol type detection, and biased UVAI retrievals over bright areas with complex terrain can increase the biases in wildfire smoke profiling.

**Table 4: Statistical evaluation of satellite-derived *SPH* products against WCL-determined *SPH* observations. Note that the satellite *SPH* information is only shown in one column to be compared with two distinctive WCL-determined *SPH* definitions. *STD* – standard deviation; *MB* – mean bias; *Q25* – lower quartile, 25% of the data lie below this value; *Q50* – median, 50% of the data lie below this value; *Q75* – upper quartile, 25% of the data lie above this value; *MAE* – mean absolute error; *RMSE* – root mean square error; *r* – Pearson's correlation coefficient score.**

| | WCL-Determined *SPH* | |
| --- | --- | --- |
| | $SPH^{top}$ | $SPH^{ext}$ |
| **MODIS-Terra/MAIAC** | | |
| # Collocated Pairs (spatial average) | 163 | |
| Lidar Observations Mean ± 1 *STD* (km) | 2.162 ± 0.542 | 1.382 ± 0.368 |
| Satellite Retrievals Mean ± 1 *STD* (km) | 0.733 ± 0.447 | |
| Lidar Observations $^{Max}$/$_{Min}$ (km) | $^{3.903}$/$_{1.254}$ | $^{2.253}$/$_{0.800}$ |
| Satellite Retrievals $^{Max}$/$_{Min}$ (km) | $^{2.114}$/$_{0.015}$ | |
| Lidar Observations *Q25*, *Q50*, *Q75* (km) | 1.776, 2.064, 2.508 | 1.131, 1.298, 1.581 |
| Satellite Retrievals *Q25*, *Q50*, *Q75* (km) | 0.438, 0.687, 0.903 | |
| *MB* (km) | -1.429 | -0. 650 |
| *MAE* (km) | 1.429 | 0.673 |
| *RMSE* (km) | 1.591 | 0.822 |
| *r* | 0.008 | **0.247 ** |
| **MODIS-Aqua/MAIAC** | | |
| # Collocated Pairs (spatial average) | 114 | |
| Lidar Observations Mean ± 1 *STD* (km) | 2.686 ± 0.797 | 1.790 ± 0.644 |
| Satellite Retrievals Mean ± 1 *STD* (km) | 0.425 ± 0.262 | |
| Lidar Observations $^{Max}$/$_{Min}$ (km) | $^{4.215}$/$_{1.374}$ | $^{3.422}$/$_{0.800}$ |
| Satellite Retrievals $^{Max}$/$_{Min}$ (km) | $^{0.935}$/$_{0.025}$ | |
| Lidar Observations *Q25*, *Q50*, *Q75* (km) | 2.063, 2.627, 3.350 | 1.274, 1.728, 2.325 |
| Satellite Retrievals *Q25*, *Q50*, *Q75* (km) | 0.192, 0.379, 0.697 | |
| *MB* (km) | -2.261 | -1.365 |
| *MAE* (km) | 2.261 | 1.365 |
| *RMSE* (km) | 2.393 | 1.525 |
| *r* | **0.219 *** | 0.057 |





| **MISR/MERLIN** | | |
|---|---|---|
| # Collocated Pairs (spatial average) | 90 | |
| Lidar Observations Mean ± 1 *STD* (km) | 2.216 ± 0.506 | 1.498 ± 0.449 |
| Satellite Retrievals Mean ± 1 *STD* (km) | 2.124 ± 0.625 | |
| Lidar Observations $^{Max}/_{Min}$ (km) | $^{2.982}/_{1.254}$ | $^{2.253}/_{0.853}$ |
| Satellite Retrievals $^{Max}/_{Min}$ (km) | $^{3.029}/_{0.625}$ | |
| Lidar Observations *Q25, Q50, Q75* (km) | 1.791, 2.204, 2.648 | 1.129, 1.428, 1.969 |
| Satellite Retrievals *Q25, Q50, Q75* (km) | 1.658, 2.083, 2.801 | |
| *MB* (km) | -0.092 | 0. 626 |
| *MAE* (km) | 0.368 | 0.719 |
| *RMSE* (km) | 0.554 | 0.788 |
| *r* | **0.551** ** | **0.648** ** |
| **VIIRS/ASHE** | | |
| # Collocated Pairs (matched pair) | 130 | |
| Lidar Observations Mean ± 1 *STD* (km) | 2.823 ± 0.999 | 1.895 ± 0.890 |
| Satellite Retrievals Mean ± 1 *STD* (km) | 2.707 ± 1.165 | |
| Lidar Observations $^{Max}/_{Min}$ (km) | $^{5.493}/_{1.497}$ | $^{4.003}/_{0.811}$ |
| Satellite Retrievals $^{Max}/_{Min}$ (km) | $^{4.930}/_{0.231}$ | |
| Lidar Observations *Q25, Q50, Q75* (km) | 1.977, 2.904, 3.318 | 1.094, 1.629, 2.489 |
| Satellite Retrievals *Q25, Q50, Q75* (km) | 2.060, 2.683, 3.579 | |
| *MB* (km) | -0.116 | 0.812 |
| *MAE* (km) | 1.190 | 1.594 |
| *RMSE* (km) | 1.697 | 1.806 |
| *r* | **-0.220** * | **-0.220** * |
| **TROPOMI/ALH** | | |
| # Collocated Pairs (matched pair) | 127 | |
| Lidar Observations Mean ± 1 *STD* (km) | 2.677 ± 1.075 | 1.894 ± 0.936 |
| Satellite Retrievals Mean ± 1 *STD* (km) | 2.052 ± 0.588 | |
| Lidar Observations $^{Max}/_{Min}$ (km) | $^{5.493}/_{1.374}$ | $^{4.003}/_{0.734}$ |
| Satellite Retrievals $^{Max}/_{Min}$ (km) | $^{3.425}/_{1.412}$ | |
| Lidar Observations *Q25, Q50, Q75* (km) | 1.718, 2.337, 3.308 | 1.019, 1.542, 2.684 |
| Satellite Retrievals *Q25, Q50, Q75* (km) | 1.546, 1.802, 2.431 | |
| *MB* (km) | -0.625 | 0.158 |
| *MAE* (km) | 0.832 | 0.847 |
| *RMSE* (km) | 1.304 | 0.991 |
| *r* | 0.151 | **0.241** ** |

*The Pearson's correlation coefficient scores (r) in bold demonstrate a statistically significant relationship between the two variables; the symbol \* signifies a p value < 0.05 and \*\* indicates a p value < 0.01. A lower p-value suggests a higher level of statistical significance.*




### 4.4 Qualitative evaluation of satellite-derived *SPH* and physical considerations

The comparison of satellite-based *SPH* with two distinctive *SPH* definitions determined by WCL data poses the following question: what other factors must be considered? To address this question, we specifically analyze factors, such as "near-fire-event region (distance from the fire source < 20 km)" or "downwind region (distance from the fire source > 20 km)", and "low AOD" (AOD < 1) or "high AOD" (AOD > 1). We then investigate the relationship between these factors for each satellite dataset (**Fig. 6**).

For MODIS-Terra/MAIAC, $SPH^{ext}$ can effectively interpret a majority of retrievals not only for the near-fire-event region but also for the downwind region, and all are not sensitive to significant variability in aerosol loading. Furthermore, the MAIAC PIH algorithm underestimates *SPH* with increasing AOD in the downwind region. Instead, for MODIS-Aqua/MAIAC, the retrievals are generally highly biased, with only a few points falling within the region between the 1:1 and 1:2 lines. In other words, the MODIS-Aqua/MAIAC data exhibits relatively more consistency with the definition of $SPH^{ext}$ near the fire source during high-AOD conditions (when AOD > 1). In terms of MISR/MERLIN, a fair agreement is observed between the retrievals and the definition of $SPH^{top}$ for the area both in the vicinity of fire and far downwind from the fire, with outliers potentially arising from thin plumes (for small AOD < 0.8) having unclear boundaries near the fire source. This is because the MISR-based automated stereoscopic image requires distinct plume-like features to provide the complete vertical profile of the smoke plume. The VIIRS/ASHE product matches better with $SPH^{top}$ other than $SPH^{ext}$. Another interesting finding is that irrespective of AOD values, the ASHE algorithm tends to overestimate *SPH* for the near-fire-event region, while underestimating *SPH* for the downwind region. Likewise, the TROPOMI/ALH product displays lower *SPH* values far away from the fire but higher ones when close to the fire, regardless of the chosen *SPH* definition and AOD conditions. The definition of $SPH^{top}$ proves useful to evaluate the TROPOMI/ALH data within the near-fire-event region if the large outliers were removed, whereas the use of $SPH^{ext}$ is more appropriate for the downwind region.

Overall, this analysis sheds light on the factors influencing the comparison between satellite-derived *SPH* and lidar-determined *SPH* definitions. These findings can aid in refining the comprehension and interpretation of *SPH* data collected from various satellite datasets. Additionally, the physical interpretation of the potential biases in the satellite *SPH* algorithms can help design future field campaigns that provide data sets for evaluation and algorithm development.



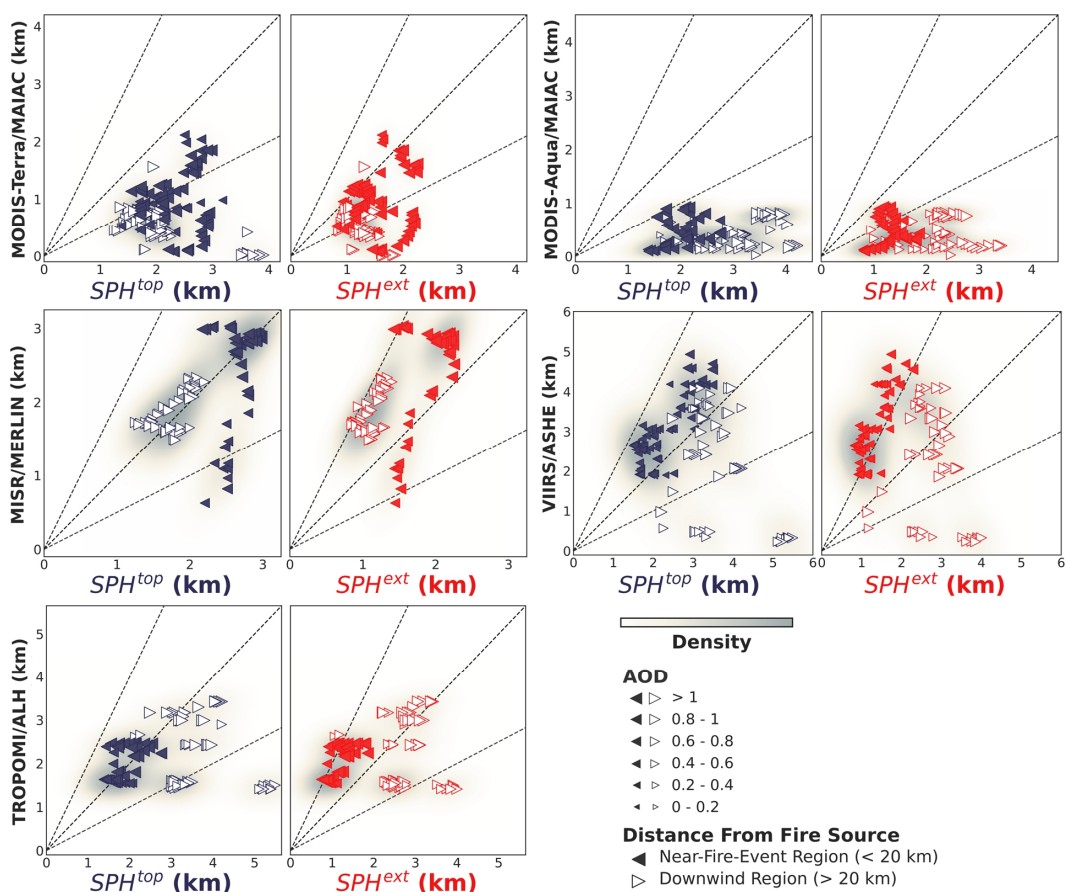

**Figure 6: Scatterplots of satellite *SPH* retrievals from MODIS-Terra/MAIAC, MODIS-Aqua/MAIAC, MISR/MERLIN,**
**VIIRS/ASHE, and TROPOMI/ALH versus the WCL-determined *SPH* using two different definitions, i.e., *SPH^top* (left, blue)**
**and *SPH^ext* (right, red), for the total collocated pairs between satellite retrievals and lidar observations from the**
**reconstructed WCL vertical cross-section in August 2018. Dotted lines denote the ratios of 2:1, 1:1, and 1:2 for reference.**
**The shaded areas show the density estimates of the overall collocated pairs' distribution. Points closer to the fire occurrence,**
**within 20 km, are shown as left-pointing triangles, while those farther away, in the downwind area, are shown as right-**
**pointing triangles. Each triangle's size can be used to infer its matching AOD value. Note: the plot axes' scales for each**
**satellite product are different.**

## 5 Summary and conclusions

The notion of *SPH* can be visualized as the vertical displacement from the ground to the upper atmosphere, marking
the extent to which smoke plumes ascend. This parameter is vital for simulating the initial stage of plume production
and predicting the potential spread of smoke from wildfires (e.g., Walter et al., 2016; Tang et al., 2022). As smoke
crosses over the PBL, it tends to persist longer and may travel farther. Within the *PBLH*, the adverse impact of smoke
on air quality can be amplified.

Current efforts to study wildfire *SPH* mainly rely on the use of active lidar data and passive satellite sensor retrievals.



However, these instruments face inherent spatial and temporal limitations, such as their inability to swiftly adapt to
changes in fire behavior. Nonetheless, fusing multi-satellite products to quantify *SPH* and prove the existing aerosol
layers is still an evolving field. Though transported smoke aerosols may form complex, multilayer structures, this
study has shown that a single, uniform aerosol layer is encountered more frequently than previously assumed, making
it possible to describe the height of the aerosol layer using a single numeric number. Scientists can more readily
incorporate aerosol layer data into climate and AQ models with this more straightforward representation, and a concept
of "effective *SPH*" is further discussed. We use two *SPH* definitions for comparisons, since the *SPH* criterion varies
between plume rise retrieval algorithms, given their diverse representations of aerosol vertical allocation that may not
sufficiently reflect the real wildfire-associated smoke aerosol layering. Then we employ two different collocation
methods to deal with the lidar-satellite collocated pairs. The collocation uncertainties can be caused by the discrepancy
between the coarse spatial resolution of the satellite retrieval algorithm and the fine-scale variability of wildfire smoke
plume activity detected by high-resolution active lidar measurements.

Results in this paper reaffirm that uncertainties in various satellite-derived *SPH* products arise from different remote
sensing techniques (Tosca et al., 2011; Flower and Kahn, 2017). The current state of satellite-based *SPH* data is
impacted by significant errors, which we ascribe mostly to either complex, multiple aerosol layers or thin, transparent
plumes. The user recommendations and main conclusions drawn from this study are:

(1) The MAIAC PIH algorithm necessitates careful quality verification since its *SPH* retrievals are routinely lower
than the lidar measurements, especially for MODIS/MAIAC-Aqua. We suggest selecting $SPH^{ext}$ as a suitable height
metric to evaluate the MODIS/MAIAC-Terra product under conditions when distance from the fire source < 20 km
and AOD at 355 nm > 1.

(2) The MISR plume height climatology is promising to help locate wildfire-associated $SPH^{top}$ and provide the best
estimates over mountainous terrain. However, as WUS fires have become more frequent since the 2000s, the available
MISR/MERLIN datasets are relatively minimal. The most striking problem is that the MISR observations can only be
made in the late morning and require labor-intensive operation of the MINX software to digitize the smoke plumes.

(3) Both the VIIRS/ASHE and the TROPOMI/ALH products show great potential for characterizing $SPH^{top}$ in a single
homogenous aerosol-rich layer. An overestimation of *SPH* in the near-fire-event region and an underestimation of
*SPH* in the downwind region still prevail in AOD of different size bins. We find that large retrieval errors occur in the
studied cases, underscoring the need for a robust quality screening approach related to the UVAI parameterization.

However, the performance evaluation of four satellite *SPH* datasets presented here indicates only a weak to moderate
correlation between passive satellite retrievals and airborne lidar observations. Deploying both passive and active
sensors in tandem can offer a synergistic approach, filling gaps in our understanding of fire and smoke plume behavior
by utilizing the unique strengths of each method. The lack of synchronization between satellite overpass times and
fire activity and aerosol layering is responsible for more than half of the collocated mismatches. It is expected that
future satellites equipped with active or passive sensors can increase the chances of capturing a large wildfire event at



its peak increase (i.e., increased temporal coverage). Notably, NASA's forthcoming aerosol investigations from space, such as ACCP (Aerosol and Cloud, Convection and Precipitation), MAIA (Multi-Angle Imager for Aerosols), PACE

(Plankton, Aerosol, Cloud, ocean Ecosystem), and TEMPO (Tropospheric Emissions: Monitoring of Pollution), are expected to play a pivotal role in this regard. By integrating data from multiple satellite systems as a potential solution to the synchronization issue, scientists can create a more comprehensive and improved picture of wildfire plume rise.

This study provides a preliminary comparison reference for multiple satellite-based *SPH* data applications. Our findings serve to connect smoke transport and AQ forecasting frameworks and future satellite missions that aim to

655 quantify the vertical distribution of aerosols in the atmosphere, similar to the efforts of Raffuse et al. (2012), Solomos et al. (2015), Ke et al. (2021), and Kumar et al. (2022). We therefore encourage conversations between the communities involved in satellite remote sensing and atmospheric modeling to enhance the diversity of perspectives and foster a consensus on the measurement and comprehension of effective *SPH* with greater clarity.

### Appendix A. Evaluation metrics for lidar-satellite collocation

We evaluate the performance of a satellite *SPH* product with respect to lidar observations using the following statistics: the mean bias (*MB*), the mean absolute error (*MAE*), the root mean square error (*RMSE*), the Pearson's correlation coefficient score (*r*). The metrics are calculated for *SPH* using the Eqs. (A1) to (A4):

$$MB = \overline{SPH_{lidar}} - \overline{SPH_{satellite}}, \tag{A1}$$

$$MAE = \frac{\sum_{i=1}^{N} |SPH_{lidar,i} - SPH_{satellite,i}|}{N}, \tag{A2}$$

$$RMSE = \sqrt{\frac{\sum_{i=1}^{N} (SPH_{lidar,i} - SPH_{satellite,i})^2}{N}}, \tag{A3}$$

$$r = \frac{\sum_{i=1}^{N} (SPH_{lidar,i} - \overline{SPH_{lidar}})(SPH_{satellite,i} - \overline{SPH_{satellite}})}{\sqrt{\sum_{i=1}^{N} (SPH_{lidar,i} - \overline{SPH_{lidar}})^2} \sqrt{\sum_{i=1}^{N} (SPH_{satellite,i} - \overline{SPH_{satellite}})^2}}, \tag{A4}$$

, where $SPH_{lidar,i}$ is the $i^{th}$ collocated lidar measurement, $SPH_{satellite,i}$ is the $i^{th}$ collocated satellite observation, $\overline{SPH_{lidar}}$ is the arithmetic mean of collocated lidar measurements, $\overline{SPH_{satellite}}$ is the arithmetic mean of collocated satellite observations, *N* is the number of collocated pairs.

*MB* represents the average bias of a satellite *SPH* product but should be interpreted cautiously because positive and negative errors will cancel out. *MAE* measures the average over the sample absolute differences between lidar measurements and satellite observations where all individual differences have equal weight, without considering their direction. *RMSE* is the square root of the average of squared differences between lidar measurements and satellite observations. *RMSE* should be more useful when large outlier errors are particularly undesirable. Unlike *RMSE*, *MAE*

is an unambiguous measure of average error magnitude. *r* is a measure of the strength of a linear association between



two variables, indicating that the spatial distribution of both lidar measurements and satellite observations for *SPH* has a similar change trend. The best performance that a satellite *SPH* product would have for these evaluation metrics is: *MB* (km) = 0, *MAE* (km) = 0, *RMSE* (km) = 0, and *r* (unitless) =1.

*Data availability.* The MODIS/MAIAC MCD19A2 Version 6.1 data product can be found at https:// earthdata.nasa.gov, last access: 10 May 2023. The Atmospheric Sciences Data Center hosts a web-based interface for freely downloading the MISR/MERLIN plume files at https://l0dup05.larc.nasa.gov/merlin/merlin#, last access: 10 August 2022. The TROPOMI/ALH Level 2 data are publicly available to users via Copernicus Open Access Hub at https://scihub.copernicus.eu/, last access: 9 February 2023. The VIIRS/ASHE data can be obtained from the VIIRS Deep Blue Aerosol Group (https://deepblue.gsfc.nasa.gov/, last access: 28 July 2022). The BB-FLUX WCL data can be obtained from the official UWKA project website (http://www.atmos.uwyo.edu/uwka/projects/index.shtml, last access: 31 October 2022). Balloon sounding data are available from Atmospheric Soundings Wyoming Weather Website (https://weather.uwyo.edu/upperair/sounding.html, last access: 22 May 2023).

*Author contribution.* J.H.: Conceptualization; Investigation; Methodology; Lidar and satellite data curation; Software; Data visualization and analysis; Writing - original draft; Writing - review & editing. S. M. L.: Investigation; Methodology; Writing - review & editing. M.D.: Methodology; Lidar data curation; Writing - review & editing. J. L.: Methodology; Satellite data curation; Writing - review & editing. H. A. H.: Conceptualization; Investigation; Methodology; Supervision; Funding acquisition; Writing - review & editing.

*Competing interests.* The authors declare that no known conflicts of interest, either financial or interpersonal, could have appeared to influence the work reported in this paper.

*Acknowledgements.* This material is based on work supported, in part, by the National Science Foundation (NSF) Chemical, Bioengineering, Environmental, and Transport Systems (CBET) under grant number 2048423. We acknowledge high-performance computing support from Cheyenne (doi:10.5065/D6RX99HX) provided by NCAR's Computational and Information Systems Laboratory, sponsored by the National Science Foundation. We thank the University of Wyoming King Air team for the successful deployment of the 2018 BB-FLUX project (PI: Rainer Volkamer). We acknowledge the use of imagery from the Worldview Snapshots application (https://wvs.earthdata.nasa.gov, last access: 15 June 2023), part of the Earth Observing System Data and Information System (EOSDIS). We appreciate the MODIS Adaptive Processing System (MODAPS) Team, the MISR Wildfire Smoke Plume Height Project, and the VIIRS Deep Blue Aerosol Group from NASA, and the TROPOMI Algorithm Team from the ESA for their efforts to create and maintain the satellite data records used in this paper.



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
