# Peer review of "Assessment of smoke plume height products derived from multisource satellite observations using lidar-derived height metrics for wildfires in the western US"

_EGUsphere, 2023_

## Author Comment (AC1)

We appreciate two anonymous referees for their constructive suggestions, which we believe have improved the overall quality of the revised manuscript (with tracked changes). We have addressed all concerns from the reviewers with additional supporting analyses and clarifications in the text. We respond below by starting with a summary of the specific changes to our first submission. Our point-to-point responses (to each comment raised by the referees) are in black with manuscript changes in blue, and referees' comments are shown in italics.

**General summary**

Here is a summary of the specific changes to the original manuscript:

1. We modified and restructured the text to improve the clarity of the manuscript and provided legible Track Changes in the resubmission so that the reviewers could see the detailed edits.
2. We corrected the typographical and grammatical errors as stated in point-by-point responses to the reviewers' comments below.
3. We changed the color schemes used in figures and charts to allow readers with color vision deficiencies to interpret our findings correctly.
4. We reorganized the "Result and Discussion" section (Sect. 4).

**Referee #1**

*> Comment 1: In this study, the authors compare upward facing lidar measurements of smoke plume characteristics from an aircraft campaign with plume height products derived from passive satellite remote sensing. To do so, they first establish two definitions of plume height, a plume top and an extinction-weighted smoke mean plume height. This is important because what is meant by plume top or plume injection height is not clear in the literature and differs by satellite product.*

*This work is a useful contribution to the field and is generally rigorous. While this topic is complex and multidimensional, the authors make it somewhat more difficult to understand with overly complex language and a few non-standard graphical choices. The lack of CALIPSO data in the study is surprising and should be explained. Otherwise, most of the comments below address clarity.*

We thank the reviewer for the positive feedback regarding our approach. We have rewritten the methodology and clarified our results by updating the figures and tables.

*> Comment 2: Line 90: This is the only mention of CALIPSO and CATS in the manuscript. The authors should address why they were not included in the analysis. The narrow swath is not justification enough, and the lack of diurnal variation is shared by many of the passive products included in the analysis. If there were not enough coincident overpasses to make meaningful statistics, that would be reason to exclude them, and would highlight the difficulty of using the spaceborne lidar products in direct comparison with aircraft campaigns.*

We agree that including CALIPSO or CATS data for comparison would be valuable to show the sensitivity of such space-borne lidars to smoke plume height (*SPH*); however, we could not find any valid pairs between the reconstructed flight tracks and CALIPSO overpasses by using our rigorous collocation requirements (search radius: 6km, and search time window: 12 minutes). Moreover, the operation time of CATS was from 10 February 2015 to 30 October 2017, which is out of our study time period. We agree that we did not explicitly explain the lack of CALIPSO and CATS in our study. To justify our choice, we added the relevant text on **lines 118–120**, **page 4** of the revised manuscript as follows:

"It is important to note here that there were no coincident satellite-based lidar overpasses for our field campaign data; therefore, they are not included in our results. This omission

underscores the difficulties in directly comparing spaceborne lidar products with data from aircraft campaigns."

> *Comment 3: Line 98: The word "hence" here is not needed because this sentence does not follow from the previous. There are many such small grammatical issues in the draft. I have identified a few, but certainly not all. The manuscript should be reviewed by an experienced technical editor.*

We thank the reviewer for bringing to our attention those grammatical errors that has been corrected throughout the revised manuscript.

> *Comment 4: Line 147: Take out "high precision." It is not defined and unclear why the MISR retrievals are higher precision than the other sensors.*

Corrected.

> *Comment 5: Line 170: Remove "subsequent."*

Corrected.

> *Comment 6: Line 274: Does this method of plume top determination work also for modeled aerosol data? It would be very useful if the same definition could be used for both model and satellite plumes. Gridded models typically exhibit exponential decay of aerosol concentrations as they go up in vertical layers. Would this calculation derive sensible SPH(top) in that case?*

Yes – the definition of $SPH_{top}$ proposed by this study is based on the wavelet covariance transform (WCT) technique, which has been widely applied in the past as one of the most reliable methods to extract geometrical features in the lofted aerosol layers from lidar data. We believe this method might be useful in deriving $SPH_{top}$ from the modeled aerosol profiles, such as the MERRA-2 aerosol reanalysis data. However, whether an aerosol extinction/backscatter profile is evenly gridded in the vertical direction has a significant impact on the structure of aerosol profiles. A coarse vertical grid resolution, common in many gridded chemical transport models, may bias the derived $SPH_{top}$. This calculation can be very sensitive to variations in the vertical gradient of the aerosol extinction/backscatter profile by assigning the optimal value of the dilation parameter $a$, which requires an iterative approach to determine using retrospective

results. It is worth noting that removing the effect of cloud signals in the entire aerosol profiles is required to effectively identify the location of $SPH_{top}$ when using this method.

> *Comment 9: Line 308: These two paragraphs are confusing. I'd suggest you start with something like:*

*Because of the differences in spatial resolution, we developed two methods for collocating lidar measurements with satellite-derived SPH products, one for the finer resolution MODIS and MISR data and one for the coarser resolution VIIRS/TROPOMI. Then go on and explain the two methods.*

Thank you for this suggestion. To make these paragraphs clearer, we changed the flow of Sect. 3.2 in the revised manuscript. Overall, this revised structure of Sect. 3.2 systematically builds on introducing the challenges, explaining the collocation methods developed to address them, detailing the data handling, assumptions, and uncertainties, and finally presenting the collocation criteria and statistical analysis used in the study.

> *Comment 10: Line 404: "upward-facing" lidar profiles.*

Corrected.

> *Comment 11: Line 406: It is very difficult for the reader to interpret this figure for fire/plume/atmosphere characteristics because those are not given in the plot. Rather, there is a flight name and that can be cross-referenced with the details given in table 2. What I take from Figure 2b is that the difference between SPH(top) and SPH(ext) is often greater within a single plume than the differences among different plumes when comparing the same metric. This seems important!*

We apologize for the misleading text used to interpret Fig. 2b. We merged Tables 2 and 3 as a single table (i.e., Table 2 in the revised manuscript) so that the general information of wildfire cases can be cross-referenced with the flight mission details. As we reordered the "Results and Discussion" section, Fig. 2b becomes a single figure, i.e., Fig. 8 in Sect. 4.4. Fig. 8 is now modified with the modeled planetary boundary layer height (*PBLH*) to illustrate the coupled interactions between the fire and atmosphere. The panels in Fig. 8 are labeled by the fire names for readers to visualize how individual aircraft observations are related.

[Figure]

**Figure 8:** Box plots comparing the 30 min average modeled *PBLH* (grey) with the WCL-determined *SPH* using two different definitions (*SPHtop*, blue; *SPHext*, red) for the morning (shaded by green) and afternoon (shaded by yellow) flight missions. Each panel represents a single wildfire case. Upper and lower whiskers represent the 95[th] and 5[th] percentiles, respectively, while the box spans from the 25[th] percentile to the 75[th] percentile. The line inside the box represents the median (the 50[th] percentile), and the triangle indicates the mean of the range of height values. Note that the range of WCL *SPH* measurements for both morning (0820a) and afternoon (0820b) flight missions on August 20, 2018, is limited because only a small fraction of flight tracks are considered valid transects for reconstruction.

Text on **lines 688–690**, **page 31** of the revised manuscript has been modified accordingly:

"In **Fig. 8**, there is no clear single pattern for the vertical spread of the smoke plume due to the fire–atmosphere coupling and boundary-layer turbulence (Sun et al., 2009; Deng et al., 2022b). The difference between *SPHtop* and *SPHext* is often greater within a single plume than the differences across different plumes."

> *Comment 12: Line 415: Couldn't SPH(ext) also be underestimated by the lidar under conditions of optically thick plumes?*

Yes, this is correct. The presence of clouds or optically dense aerosol layers may attenuate the WCL's signal, resulting in large uncertainties in the measured profiles or missing data. We have added more explanation to explicitly describe when lidar-derived *SPH* biases should not be overlooked. The text on **lines 544–554**, **pages 20–21** of the revised manuscript has been modified to:

"It should be noted that the upward-sampled WCL can only provide a partial vertical segment and not a fully resolved cross-section of the smoke plumes from the lowest flight height due to the restricted lidar laser penetration in optically thick smoke plumes. For instance, when probing the plume centerline, there is complete attenuation of the lidar beam, resulting in a loss of data samples. However, the WCL can successfully delineate the atmosphere on each pass in the

less dense portions of smoke plumes. Therefore, the vertical structure of individual smoke plumes reconstructed from airborne WCL measurements yields the vertical profiles of the mean aerosol extinction coefficient, reflecting the average conditions of smoke plumes over multileg UWKA sampling periods (see more details in **Sects. 3.3** and **4.3**). In terms of lidar-derived $SPH$ biases identified in our study, we observe three main scenarios: (1) underestimation of $SPH_{top}$ (i.e., optically thick plumes limiting vertical extent); (2) overestimation of $SPH_{ext}$ (i.e., the upward-pointing lidar not sampling below aircraft); (3) underestimation of both $SPH_{top}$ and $SPH_{ext}$ in situations where the lidar faces both dense smoke above and cannot measure below the aircraft."

> *Comment 13: Line 425: While 1000 acres is sometimes used as a definition for a large wildfire, I don't think it makes sense in this context. First, your data set consists of fires that are all at least an order of magnitude larger than that. Also, is there evidence of many fires of ~1000 acres that produced plumes lofted to the free troposphere or stratosphere?*

We appreciate the reviewer's point regarding the use of "1000 acres" as a benchmark for a large wildfire. We agree that this definition may not be the most fitting for our study, as the fires we analyzed are larger than 10,000 hectares (~24,700 acres). This leads us to align more accurately with the term "megafires", as characterized in Linley et al. (2022), which suggests the term "megafire" for a single fire greater than 10,000 hectares and provides two additional thresholds/terms for even larger fires "gigafire" and "terafire".

Furthermore, we acknowledge the variability in the literature regarding the terms "large fire", "very large fire", "extremely large fire", and "megafire" which lack a uniform definition. To better reflect the nature and scale of the fires in our study, we propose to replace the term "large" with "extremely large" in our context. This choice of wording not only addresses the size of the wildfires but also incorporates other defining characteristics, such as their intense behavior and resistance to control, as highlighted by Tedim et al. (2018).

Text has been added with references, on **lines 676–680**, **page 30** of the revised manuscript:

"However, large wildfires can have vigorous buoyant plume cores that lift the smoke plume into the free troposphere (FT) or even the stratosphere (Fromm et al., 2019) contributing to elevated aerosol concentrations above the $PBLH$. Based on burned area in **Table 2**, the fires in our study meet the definition of a megafire (10,000–100,000 ha) suggested by Linley et al. (2022),

but it should be noted that fire size alone cannot characterize the fire intensity or activity and the resulting smoke plume behavior (Tedim et al., 2018)."

> *Comment 14: Line 460: I find this plot very hard to decode. The use of the negative and positive x-axis to distinguish morning and afternoon makes it difficult to compare morning/afternoon pairs directly. Also, the use of flight codes instead of fire names makes it hard to know how individual observations are related. Your readers were not on the flight campaign. I would suggest using multiple panels, with each panel representing a single fire. Use color or location to indicate morning vs. afternoon and put all heights on the same axis.*

Following the reviewer's suggestion, we removed this plot (Fig. 3 in the original manuscript) from the revised manuscript and merged it into Fig. 2b, which was presented in Sect. 4.1 of the original manuscript. Please refer to the modified plot, Fig. 8 in Sect. 4.4, to see the relevant changes.

> *Comment 15: Line 531: Remove the word "activity."*

Corrected.

> *Comment 16: Line 599: Figure 6 seems to be punchline and most important part of the study. I wish it had come sooner in the paper. I would put it at the beginning of the results section.*

Fig. 6 was presented in Sect. 4.4 of the original manuscript. Taking this comment into account, we decided to move this plot up as Fig. 4 at the beginning of the "Results and Discussion" section (Sect. 4.1). Note that in the revised version of the manuscript, Sects. 4.3 (Quantitative Evaluation) and 4.4 (Qualitative Evaluation) in the original manuscript become a single Sect. 4.1.

> *Comment 17: Line 618: Remove the word "numeric."*

Corrected.

**Referee #2**

> *Comment 1: This study seeks to first provide standard definitions of lidar-derived smoke plume height, the plume top height and the effective plume height, which is based on an extinction-weighted mean. Then, 4 different smoke plume height retrieval algorithms based on passive space-borne remote sensing instruments are evaluated against each of the lidar smoke plume height metrics in order to determine which metric is a better evaluator for each product.*

*I believe the core science is good and worthy of publication, however I feel that numerous issues detract from the focus and in the end will leave readers somewhat confused. My comments and suggestions are described below, which generally relate to conciseness and clarity.*

We thank the reviewer for the insightful comments. The suggestions given by the reviewer helped to clarify our work and improve the quality of the paper. We have made significant revisions to address the reviewer's concerns.

*Major comments:*

> *Comment 2: **Please include more background information on the errors and biases (including sampling bias) of the four algorithms.***

*For example, you mention that the MODIS/MAIAC algorithm has a maximum plume height of 10km and requires $AOD_{470} > 0.8$. I'd like to see some brief discussion (w/ references) on how these requirements limit available smoke plume sample, i.e. what percentage of smoke plumes is MODIS/MAIAC useable for? I realize that the point of this paper is a direct comparison of all of these products, but background information on previous validation/verification of these products is still required. While comparisons of MAIAC and ALH with MISR and CALIOP are mentioned, I'd like more and with quantitative discussion included (Side note – if the cited studies involving comparisons with MISR are in fact with the same MERLIN product used here [admittedly I do not know], then even \*more\* detail should be provided). The same goes for MERLIN and ASHE – no mention of retrieval uncertainty, errors, or biases from literature. This is critical information when comparing four different retrieval methods from different sensors, especially considering that each of the instruments may just be "seeing" different things differently (i.e. the fundamental differences between a weighted mean height based on extinction at a given wavelength described for the lidar data in the study vs passive-based methods that fundamentally rely on some effective emission height of the aerosol layer at various wavelengths, not even getting into the homogeneity of the*

*smoke layers microphysical properties). I think table 1 would be a good place for some of this information (at least the given retrieval uncertainties).*

We acknowledge the importance of providing background information on each passive satellite technique included in our study for better interpretation of our results. In response to your comment, we have expanded Table 1 to include two additional columns: "retrieval method" and "references".

None of the *SPH* products provide retrieval uncertainties; therefore, we cannot provide those here. In terms of adding details regarding previous verification/validation studies, previous studies often relied on limited case studies and simplified assumptions, making it difficult to generalize retrieval uncertainties. Also, many of these products are underused in the scientific community; therefore, comparison studies with ground-based lidar and aircraft lidar are limited. Throughout Sects. 2.1.1 to 2.1.4, we have attempted to add more details related to previous comparative analyses. We hope that the enhancements to Table 1 and our expanded discussion in the text offer a clearer and more comprehensive understanding of the capabilities and limitations of each passive satellite retrieval method used in our study.

> *Comment 3:* **Section 3.2 is very confusing and needs to be rewritten.**

*Explaining precisely how collocation between multiple platforms is done is paramount, even more so when comparing specifically in situ and satellite data. This section seemed very out-of-order to me. The first paragraph is good. After that, start simply – tell me which dataset you are starting from (i.e., aircraft or satellite), the temporal and spatial ranges for collocation, and then how multiple matches are handled. Then go into caveats, exceptions, etc. A figure showing an example of how collocation is done, especially given that there are two collocation methods used, should be included.*

This point is also discussed in the response to Reviewer #1, Comment 9 above. Following this comment, we also provided a conceptual diagram (Fig. 3 in the revised manuscript) to show the two collocation methods used in our study.

[Figure]

**Figure 3: Conceptual diagram of two collocation methods used in our study to pair aircraft observations and passive satellite retrievals. Our collocation criteria are a search radius of 6 km and a sampling time window of 12 minutes.**

> *Comment 4:* **The discussion on boundary layer height within section 4.1 seems out of place in this paper** *and distracts from the focus, which is largely the comparison between the SPH products with the aircraft lidar. Especially considering you only went halfway with the modeling and did not use a finer resolution coupled fire/atmosphere model (I understand why you would not want to – it's well beyond the scope here). The portion on page 16 in particular just seems to meander without much of a point. Either tighten it up by justifying why it's there and explicitly describing what we've gained by that portion of the work or remove it.*

Thank you for the suggestion. We have decided to shift this discussion to the end of the "Results and Discussion" section (now Sect. 4.4). This part is important to illustrate the concepts of our two proposed standard *SPH* definitions, especially when compared with *PBLH*. The application of the ratio of *SPH* to *PBLH* is also critical in air quality studies. This ratio serves as an indicator of the percentage of smoke aerosols retained below the *PBLH* that are the primary contributors to high particulate matter concentrations near the ground. A higher *SPH:PBLH* suggests a high probability of transported smoke plumes from smoke aerosols being injected above the *PBLH* (i.e., long-range smoke transport), thereby affecting extensive areas downwind. We present these results as a practical application for *SPH*. We are looking forward to other researchers joining in such a conversation to conduct advanced research using satellite-derived products to calculate this ratio.

> *Comment 5: Also, writing can be improved, most often by simplifying - e.g. "…make it work…" on line 80 is too informal, "…with high precision…" on line 147 is unnecessary, Line 239 "It has been recognized…" can be changed to "It is…" and the rest of the sentence adjusted accordingly. Line 451 "It was accidentally begun by a crashed helicopter…" is unnecessary. These are only a few examples. Please revise the manuscript with conciseness, phrasing, and sentence structure in mind.*

We appreciate the reviewer for the guidance in enhancing our manuscript. We have made several changes throughout the paper to improve clarity and conciseness, including the specific examples mentioned.

*Minor comments:*

> *Comment 6: Line 43: I'm assuming based on the context following that by "remote sensing" you mean specifically passive remote sensing, because space-borne active sensor such as CALIOP can (could?) retrieve a layered height could it not? Please be specific. Also, if you \*are\* referring to passive remote sensing, please change "observations" on line 44 to "retrievals" as passive remote sensing instruments do not directly observe heights.*

Yes, the language use of "remote sensing" in our original manuscript means "passive remote sensing". We have specified the text as suggested. We also changed the text to use "satellite retrievals" to indicate that passive remote sensing instruments do not directly observe heights.

> *Comment 7: Line 80: This sentence is poorly constructed.*

We rephrased this sentence following the reviewer's comment on **line 82**, **page 3** of the revised manuscript:

"In a subsequent study, Lee et al. (2020) revised the ASHE algorithm to function without the lidar backscatter profile."

> *Comment 8: Lines 87-90: I'm not quite sure why you just from SPH retrieval to fire detection – these are fundamentally different remote sensing problems.*

Yes, the reviewer is correct. **Lines 89–92**, **page 3** of the revised manuscript have been modified accordingly:

"Passive satellites excel by delivering widespread coverage on a regular basis, all while incurring minimal recurring costs and posing no risks to observers. Yet, dense smoke plumes, cloud cover, or scan gaps between adjoining orbits of sun-synchronous polar satellites can result in unsuccessful retrievals (Lyapustin et al., 2008)."

> *Comment 9: Line 110: "…of which SPH definition can effectively interpret a specific satellite SPH retrieval algorithm" is confusing, perhaps due to word choice.*

We rephrased this sentence following the reviewer's comment, on **lines 111–112**, **page 3** of the revised manuscript:

"The primary objective of this study is to address the central research question: which *SPH* definition corresponds to the most physically relevant plume height for a specific satellite *SPH* retrieval algorithm?"

> *Comment 10: Table 1. Please include the proper references for each algorithm within the table.*

Thanks for the helpful comment. We have added relevant references with respect to each satellite *SPH* retrieval algorithm in Table 1.

> *Comment 11: Line 133: What exactly is unique about MODIS' ability to detect fires?*

Direct broadcast mode is one of the unique features of the MODIS instrument, which can perform near-real-time fire monitoring. Also, the twin set of MODIS on board the Terra and Aqua satellites offered a chance starting in the early 2000s to improve the ability to capture changes in fire activity throughout the day, where detections occur twice daily in the mid-late morning and early afternoon.

Since our focus is on simplifying the language and improving sentence structure throughout the text, we have removed unnecessary descriptions from **lines 135–137**, **page 5** of the revised manuscript:

"MODIS sensors are located on the Terra (morning sensor, 10:30 AM local solar time) and Aqua (afternoon sensor, 1:30 PM local solar time) satellite platforms and operate in the TIR spectrum to detect active fires (Salomonson et al., 2002)."

> *Comment 12: Line 150: You mention "…wealth of data collected…over two decades…" here, which is contradictory to the data availability of the MISR listed in the "time period" column of table 1.*

To clarify, the sentence in question refers to the data collection duration of the MISR instrument, which indeed spans over two decades. However, the online availability of post-processed MISR/MERLIN wildfire plume height products, as listed in Table 1, is limited to the years specified. This arises from the necessity to manually process the *SPH* retrievals, including the delineation of smoke plume regions and data extraction. We hope this explanation addresses this concern. We modified the relevant text on **lines 154–161**, **page 6** of the revised manuscript as follows:

"The wealth of data collected by the MISR instrument over two decades offers valuable insights into the global climatology of fire in the environment, across geographic regions, biomes, and seasons (Val Martin et al., 2018; Gonzalez-Alonso et al., 2019). The publicly available database built using manually postprocessed MISR products has been used to evaluate plume rise models (e.g., Ke et al., 2021) and other satellite-derived datasets (e.g., Lyapustin et al., 2019; Griffin et al., 2020). Recently, an interactive visualization tool called MERLIN was developed to facilitate the exploration and accessibility of over 70,000 records of global wildfire plume height retrievals (Boone et al., 2018; Nastan et al., 2022)."

> *Comment 13: Line 175: There are now 3 VIIRS sensors in space – why is the revisit time listed here as 12 hours? (I see now this is because of the field campaign limiting the study period. This information should be mentioned, if not fully provided, to prevent confusion. This requires moving table 2 up.)*

A footnote has been appended to Table 1, clarifying that the Suomi NPP VIIRS aerosol observation is used in the current version of the ASHE algorithm to derive *SPH* results. It is important to note that we anticipate the extension of this retrieval algorithm to other VIIRS instruments in the future. Furthermore, to address potential confusion and provide context as suggested, we have included additional explanation in the revised manuscript on **lines 201–207**, **page 7**:

"Preliminary evaluation suggested that the ASHE-retrieved *SPH* had an uncertainty of 1–1.2km (or 30–40% for *SPH* of 3 km) for heavy aerosol loading cases (AOD > 1) (Lee et al., 2016,

2020). The uncertainty is dependent on errors in retrieved AOD, assumed aerosol optical model, and surface reflectance, and generally decreases with increasing AOD. It should be noted that OMPS-NM abroad SNPP has a relatively coarse spatial resolution of ~50 × 50 km$^2$ near nadir (~200 × 100 km$^2$ near the edge of the across-track scan), indicating that it has limitations for small-scale (subpixel) smoke plumes. Although there are multiple VIIRS instruments, the ASHE product is currently only available for SNPP VIIRS. It is anticipated that this retrieval algorithm will be implemented for other VIIRS instruments in the future."

> Comment 14: Line 220: I can guess why prescribed and small fires are excluded from this study but you do not want to leave readers guessing – please provide a sentence explaining why.

As the sentence seems unclear, we better explain it on **lines 236–240**, **page 8** of the revised version as below:

"Small fires were not included, in part, because of the expected large uncertainties in satellite retrievals of the relatively low *SPH* values (ranging from hundreds of meters for prescribed fires to thousands of meters for small fires). Large errors for smoke aerosol layers within the boundary arise from a mismatch between the coarse spatial resolution of satellite pixels and the fine-scale smoke plume variability inherent in wildfire activity (Geddes and Boesch, 2015)."

> Comment 15: Line 239: This sentence needs re-writing.

Changed. Text on **lines 260–264**, **page 10** of the revised manuscript has been modified accordingly:

"In previous studies, the aerosol extinction coefficient is one of the most frequently observed and reported aerosol optical properties to characterize the vertical structure of the atmosphere and develop a height retrieval algorithm (Gordon, 1997; Dubovik et al., 2011; Sanghavi et al., 2012; Hollstein and Fischer, 2014; Ding et al., 2016; Wu et al., 2016; Xu et al., 2017)."

> Comment 16: Line 253: I believe your use of a superscript in a parameter name ($SPH^{TOP}$) is the first I've ever seen, or at least can remember. Unless you have a reason for doing so, I recommend going with standard procedure and switching it to $SPH_{TOP}$. Same goes for $SPH^{ext}$.

We have corrected "$SPH^{top}$" to "$SPH_{top}$", and "$SPH^{ext}$" to "$SPH_{ext}$" throughout the text and updated plots.

> *Comment 17: Lines 309-313. This section is quite unclear and specifically I do not understand what you mean by saying that VIIRS/ASHE has a "temporal duration" of 6 minutes.*

According to the VIIRS Level 2 aerosol and fire product user's guide, a single VIIRS granule contains a ~6-minute orbital segment spanning multiple scans, with each individual scan containing a fixed number of rows, with one row for each detector. To clarify this point, text on **lines 364–369**, **page 14** of the revised manuscript has been modified accordingly:

"A single granule of the VIIRS/ASHE product has the largest pixel size (6 km × 6 km) with the longest orbit segment scanning period (~ 6 minutes) of all the satellite-derived $SPH$ products in **Table 1**. To ensure that adequate collocation pairs are available within one half hour due to rapid wildfire smoke plume activity, we utilized a sampling time window of 12 minutes that corresponds to twice the maximum time span of an orbital swath (one scene)."

> *Comment 18: Line 356: "fire plume"?*

The sentence has been rephrased, on **lines 388–390**, **page 14** of the revised manuscript:

"When the UWKA flew close-to-perpendicular to the mean wind direction, the consecutive UWKA transects sampled the smoke plumes at different heights over the same latitude or longitude range of the flight trajectory."

> *Comment 19: Line 359: So the WCL signal is attenuated by thick smoke, but many of the satellite retrievals require optically thick smoke layers – at what optical thickness does lidar attenuation become an issue?*

Part of this point is also discussed in the response to Reviewer #1, Comment 12 above. To clarify, the dense smoke cases with high aerosol optical depth (AOD > 1.5 at 355 nm) are where the WCL signals are strongly attenuated (personal communication with co-author Min Deng). In our study, we take advantage of the reconstructed vertical structure during wildfire events to reflect the average conditions of smoke plumes. As seen from Sect. 4.3 in the revised manuscript, satellite retrievals such as MISR/MERLIN and VIIRS/ASHE under high AOD

conditions can be comparable to $SPH_{top}$ measured by WCL; but some of them still perform poorly due to fundamental problems in satellite retrieval algorithms.

*> Comment 20: Figure 2. Include SPH^Top and SPH^ext as figure titles or at least as a label somewhere on the top two panels. Also, is it necessary to plot them separately and thus making it more difficult to directly compare them? I'd suggest doing something like merging them into one panel, removing the CDF as it's basically redundant to the PDF lines, and then using dotting/dashing to indicate the ratios. That way both methods can be shown on the same axis and compared more easily. Even though they are not really intended for comparing with each other, I still think it's important to be able to quickly visualize the differences between them and plotting them on a different axis precludes this. If sticking with two separate panels, they should be labeled "(a)" and "(b)", as they are separate panels.*

Thank you for your suggestion regarding Fig. 2. We have implemented two separate panels, now labeled as 'a' and 'b', to accommodate the distinct x-axis ranges for $SPH_{top}$ (0.5 km to 5.5 km) and $SPH_{ext}$ (0.5 km to 4.5 km). We have also removed the CDF lines to enhance clarity and placed the $SPH_{top}$ and $SPH_{ext}$ labels next to the PDF lines for easier comparison. We believe these changes will significantly improve the figure readability and facilitate a better understanding of the data. Please see the revised Fig. 5 in Sect. 4.2.

*> Comment 21: Line 425: 1000 acres is approximately equal to 4 km², which is about one singular pixel from an instrument such as MODIS. Showing acres burned in table 3 but using acres burning for the definition of "large wildfire" is confusing at first glance. Maybe change the information in table 3 to something like the maximum concurrent acres burning.*

Part of this point is also discussed in the response to Reviewer #1, Comment 13 above. The definition of large fires is based on the burned area and not on acres burning, and we have modified this in the text.

*> Comment 22: Figure 4: the orange color of the MISR/MERLIN line needs to be changed – it is extremely difficult to see against the extinction.*

We agree that the colored lines in Fig. 4 were not clear in the original version. We adjusted the color palettes to be well-suited to the characteristics of our data and our visualization goals.

Please refer to Fig. 6 in the revised manuscript to see those changes. The plots in the supplementary information with the same color scheme are modified accordingly.

> *Comment 23: Figure 5: the pink lines are too similarly colored.*

The color palettes of Fig. 5 are modified as stated above. Please refer to Fig. 7 in the revised manuscript to see those changes.

> *Comment 24: Line 480: "It is not advisable to use the MODIS-Aqua/MAIAC product for estimating downwind SPH due to its suboptimal performance in such scenarios" I think this is an overly aggressive statement based on one case.*

This statement was not based only on one case "0819b", because the cases "0804b" and "0808b" also show similar results, that the MODIS-Aqua/MAIAC product is not applicable for smoke transported downwind. Our findings are consistent with previous findings, including one study from algorithm developers (Lyapustin et al., 2019). We rephrased the sentence on **lines 615–618**, **page 24** of the revised manuscript as follows:

"We recommend caution when using the MODIS-Aqua/MAIAC product for estimating downwind *SPH*, as its effectiveness in such scenarios is not always optimal (also refer to **Figs. S4b** and **S4c** for more details). This limitation in the MAIAC PIH algorithm has also been reported in previous studies (Lyapustin et al., 2019; Loría-Salazar et al., 2021)."

> *Comment 25: Figure 6. Please include values like r2 and rmse on the panels – going back and forth between figure 6 and the previous tables is cumbersome.*

Fig. 6 is renamed to Fig. 4 as we reorganized the "Result and Discussion" section. Following the reviewer's suggestion, we added evaluation metrics to the subpanels of Fig. 4. However, Table 4 of the original manuscript is removed and additional statistics in that table become the new Appendix B in accordance with the figure's modifications.

[revised manuscript text omitted]

---

## Referee Report (RR1)

The authors have more than adequately addressed all reviewer comments therefore I don't feel it's necessary to give full comments and recommend acceptance with copy and technical edits.

---

## Author Response (AR2)

We thank the reviewer and editor for their consideration of our revised manuscript for publication. Although Reviewer #2's comments did not indicate any technical edits, we implemented minor modifications throughout the text, marked with tracked changes in red in the final version of the manuscript. These adjustments include correcting a few grammatical errors and rephrasing some sentences. Note that these changes do not alter the scientific content of the manuscript from its last revised version.

[revised manuscript text omitted]